# Photobiomodulation Meets Mechanotransduction: Immune-Stromal Crosstalk in Orthodontic Remodeling

**DOI:** 10.3390/biomedicines13102495

**Published:** 2025-10-13

**Authors:** Jovan Marković, Miodrag Čolić

**Affiliations:** 1Clinic for Orthodontics, School of Dental Medicine, University of Belgrade, 11000 Belgrade, Serbia; jovan.markovic@stomf.bg.ac.rs; 2Serbian Academy of Sciences and Arts, 11000 Belgrade, Serbia; 3Medical Faculty Foča, University of East Sarajevo, 73300 Foča, Bosnia and Herzegovina

**Keywords:** orthodontic tooth movement, photobiomodulation, cellular cross-talk, immunomodulation, bone remodeling

## Abstract

Orthodontic tooth movement (OTM) arises from force-induced mechanotransduction within the periodontal ligament (PDL), which coordinates osteoblast and osteoclast activity with immune responses to remodel the PDL and alveolar bone. This review integrates contemporary biological insights on OTM and assesses photobiomodulation (PBM) as an adjunctive therapy. We propose that mechanical and photonic inputs may interact and potentiate signaling through the Ca^2+^-NFAT, MAPK (ERK, p38, JNK), PI3K–Akt–mTOR, NF-kB, TGF-β/Smad, and Wnt/β-catenin pathways. Such interaction could influence processes such as cell proliferation, differentiation, specific cellular functions, apoptosis, autophagy, and communication between stromal and immune cells. This convergence establishes a solid foundation for understanding the context-dependent effects of PBM in OTM. In principle, PBM appears most effective as a phase-tuned adjunct, promoting early inflammatory recruitment of osteoclasts and subsequently facilitating late-phase remodeling through immunomodulatory and reparative mechanisms. However, inconsistent irradiation parameters, small sample sizes, trial heterogeneity, and the absence of mechanistic endpoints undermine current conclusions. Furthermore, the lack of integrated PBM–OTM models limits mechanistic understanding, as much of the available evidence is derived from non-OTM contexts. Overall, PBM remains a promising adjunct in orthodontics, with the potential to integrate mechanical and photonic signals in a phase-dependent manner, though its application is not yet standardized.

## 1. Introduction

Orthodontic tooth movement (OTM) is initiated by controlled mechanical loading of the teeth, leading to remodeling of the periodontal ligament (PDL) and alveolar bone through mechanotransduction. Compression promotes osteoclastogenesis, whereas tension supports osteoblast activity [1,2,3,4]. Although the underlying mechanisms are well characterized, the clinical demand to shorten treatment duration and minimize adverse effects has stimulated interest in adjunctive approaches beyond conventional orthodontic mechanics [5,6].

Low-level laser therapy (LLLT), a noninvasive form of photobiomodulation (PBM), utilizes visible red to near-infrared laser light to modulate biological responses. PBM is the broader term that encompasses LLLT as well as non-coherent sources such as light-emitting diodes (LEDs) and broad-band lamps, all of which can produce comparable biological effects [7]. In this review, the terms “PBM” and “LLLT” are used according to the terminology of the cited studies. The biological effects of PBM are highly dependent on wavelength, irradiance, energy density, pulse structure, and treatment scheduling. Consequently, heterogeneity across clinical studies has resulted in variable and often non-comparable findings regarding its influence on OTM and treatment-related discomfort [8,9].

While the integration of photonic energy and mechanical stress in OTM remains largely unexplored, we propose a phase-tuned framework in which these signals intersect at several shared molecular hubs. Through these hubs, processes such as cellular proliferation, osteoclastogenesis, extracellular matrix (ECM) remodeling, apoptosis, autophagy, inflammation, and immune regulation are modulated. In particular, immune–stromal crosstalk is pivotal, yet many aspects of this interaction remain incompletely understood.

Overall, PBM may transiently enhance the early inflammatory response and osteoclast recruitment, followed by modulation of remodeling and facilitation of resolution and repair during later phases. In this review, we summarize the available preclinical and clinical evidence regarding the signaling mechanisms through which PBM exerts its biological effects. While much of this evidence derives from non-OTM models, it nonetheless provides indirect support for the potential utility of PBM in orthodontics [7,9].

## 2. The Biological and Molecular Basis of Orthodontic Tooth Movement

To contextualize how mechanical stress and photon energy may converge to accelerate OTM, this section briefly outlines the biological and molecular basis of OTM.

OTM begins when teeth are subjected to mechanical force, triggering a cascade of cellular and molecular events that remodel the PDL and alveolar bone [10]. The process unfolds through overlapping phases rather than in a strictly linear sequence, with timing influenced by bone density, systemic health, and the magnitude and scheduling of applied forces [1,2,10].

The earliest responses occur within 48 h of force application, characterized by an aseptic inflammatory reaction due to PDL deformation. Leukocyte infiltration and mediator release accompany this phase, coinciding with a minor immediate displacement of the tooth within its socket and the onset of periodontal tissue remodeling [10,11,12]. Local cells, such as PDL fibroblasts, PDL stromal/stem cells (PDLSCs), endothelial cells, and alveolar osteocytes, sense load energy via mechanoreceptors such as integrins, primary cilia, and stretch-activated ion channels (e.g., Piezo1) [10,11].

Subsequently, a lag phase typically emerges as acute inflammation transitions to a chronic response. Leukocyte migration persists, and structural remodeling ensues. Compressed PDL regions develop hyalinized zones, which are cleared by macrophages and multinucleated phagocytes. Monocytes differentiate into osteoclast precursors under receptor activator of nuclear factor κB ligand (RANKL) signaling. Once hyalinized and necrotic zones are cleared, tooth movement accelerates, typically in weeks 2–3. After the initial attenuation, inflammation intensifies as macrophages, fibroblasts, osteoclasts, and other infiltrating cells are activated, culminating in remodeling. On the pressure side, an increased RANKL/osteoprotegerin (OPG) ratio and heightened matrix metalloproteinase (MMP) activity promote osteoclastogenesis, while acidic resorption lacunae facilitate matrix degradation [1,13,14]. In contrast, osteoblast differentiation and matrix mineralization are favored on the tension side [10,15,16].

The compression-induced microenvironment thus favors osteoclast formation. In this process, PDL fibroblasts, PDLSCs, osteoblasts, osteocytes, and T cells play a key role by increasing RANKL and macrophage-colony stimulating factor (M-CSF), which recruit osteoclast precursors and stimulate their maturation. Proinflammatory cytokines such as interleukin (IL)-1β, tumor necrosis factor (TNF), and IL-6, produced in the early inflammatory phase, additionally stimulate osteoclasts. As the process advances, RANKL remains the key stimulator for osteoclast maturation [16,17].

Osteoclastogenesis proceeds through the concurrent activation of multiple transcription factors, including nuclear factor of activated T-cells 1 (NFATc1), activator protein-1 (AP-1), nuclear factor κB (NF-κB), microphthalmia-associated transcription factor (MITF), cAMP response element-binding protein (CREB), Forkhead box O (FoxO), and others [1,2,10,11,12,13,14,15,16,17,18,19,20,21,22,23]. Table 1 summarizes the mechanosensitive receptors, ion channels, and additional receptors that are primarily activated by mechanical force during OTM.

The initial signaling triggered by these sensors is transmitted to intermediate molecules and secondary messengers. This signaling is then propagated through a complex network of interwoven pathways. The result is the stimulation of osteoclast formation and the activation of their osteolytic functions [1,12,16].

Prostaglandin E2 (PGE2), synthesized via cyclooxygenase-2 (COX-2) activity, promotes vasodilation and increases vascular permeability in the early phase of OTM [24]. As hyalinized debris clears, PGE_2_, together with hypoxia-inducible factor-1 alpha (HIF-1α), induces angiogenesis via vascular endothelial growth factor (VEGF) and other angiogenic mediators [22]. These, together with osteoclast-derived cues, also drive the induction and activation of MMPs, which degrade the ECM, including type IV collagen, thereby facilitating cell migration toward resorption sites [19].

Osteogenesis is a complex process that involves proliferation and differentiation of osteoblastic precursors, followed by ECM synthesis and maturation, ultimately leading to matrix mineralization and formation of mature bone. These processes are triggered by the same mechanoreceptors and secondary messengers as compression-induced signaling, but they activate distinct transcriptional programs. Runt-related transcription factor 2 (RUNX2), the master regulator, commits mesenchymal precursors to the osteoblast lineage, integrating bone morphogenic protein (BMP)/Smad and Wnt/β-catenin inputs to induce osteogenic genes such as collagen type I alpha 1 chain (COL1A1), alkaline phosphatase (ALP), secreted phosphoprotein 1 (SPP1) [25].

Osterix (SP7) is a RUNX2-dependent factor that drives maturation of pre-osteoblasts into matrix-producing osteoblasts, suppresses the differentiation of chondrocytes, and upregulates genes like COL1A1, integrin-binding sialoprotein (IBSP), and BGLAP (osteocalcin). Activating transcription factor 4 (ATF4) controls osteoblast function and matrix production, such as type I collagen synthesis and osteocalcin transcription. It also aids in tuning the RANKL/OPG balance that governs osteoblast–osteoclast crosstalk. Upregulation of osteoblastic transcription factors is enabled by the downregulation of glycogen synthase kinase-3β (GSK-3β), sclerostin (SOST), and Dickkopf-1 (DKK1) [18,25].

How these and other transcription factors are activated and which signaling molecules are involved is summarized in Table 1. Most pathways, including Ca^2+^/calcineurin–NFAT, mitogen-activated protein kinases (MAPKs); extracellular signal-regulated kinase (ERK), p38 mitogen-activated protein kinase (p38), and c-Jun N-terminal kinase (JNK), phosphatidylinositol 3-kinase–Akt–mechanistic target of rapamycin pathway (PI3K–Akt–mTOR), and nuclear factor κB (NF-κB), are likewise activated on the compression side. The specific signaling in osteoblasts includes the BMP family and transforming growth factor-β (TGF-β). By engaging Smad proteins, these cytokines promote osteoblast differentiation, enhance collagen synthesis, and recruit osteoprogenitor cells [1,2,10,11,12,13,18,22,26,27,28,29,30,31]. Insulin-like growth factor-1 (IGF-1) not only supports the survival of osteoblasts but also, in conjunction with BMP-2 and often in collaboration with Wnt signaling, promotes both the proliferation and maturation of these cells [26]. During consolidation, tooth movement stabilizes, with ALP synthesis increasing as bone and PDL cells continue remodeling. As mechanical forces diminish in intensity during this late phase, inflammatory activity subsequently declines and homeostasis is gradually re-established [1,3,12].

It can be concluded that the coordinated action of stromal and infiltrating immune cells in PDL and alveolar bones leads to successful tissue remodeling to allow tooth movement. These processes are orchestrated through mechanosensitive receptors, mechanotransduction signaling pathways, cytokines, growth factors, immunological cues, and receptor-ligand interactions.

**Table 1 biomedicines-13-02495-t001:** Activation of key transcription factors during orthodontic tooth movement (OTM): receptors, intermediary molecules, signaling pathways, and resulting outcomes.

Transcription Factor	Mechanosensors/Receptors	Intermediary Molecules	Signaling Pathways	Specific Outcomes	References
**Osteoclastogenesis**					
NFATc1 (master regulator)	RANK (RANKL–RANK); ITAM-coupled receptors (OSCAR, TREM2/FcRγ); αvβ3 integrin; Piezo1/TRPV4; P2X7/P2Y	Oscillatory Ca^2+^ (SOCE); ATP; ROS (NOX2); NO	RANK → TRAF6 → NF-κB/MAPKs; ITAM receptors/Syk → PLCγ2 → CaN → NFATc1 (auto-amplification)	OC: Differentiation and fusion; activation of resorptive genes (CTSK, ACP5/TRAP) and others	[1,2,16,19,20]
AP-1 (c-Fos)	RANK; TNF receptors; integrins	ROS; Ca^2+^	ERK1/2/JNK/p38 → AP-1; synergy with NFATc1	Commitment to OC lineage; survival; priming of the resorptive programme	[1,16,18,21]
NF-κB (p65/p50)	RANK; TLRs (context)	ROS; ATP; NO	IKK → NF-κB activation; cross-talk with MAPKs and NFATc1	Early differentiation; survival; inflammatory response	[1,16,19,20]
MITF	RANK; integrins; M-CSF (c-Fms)	Ca^2+^; ROS	ERK/PI3K–Akt; cooperation with NFATc1	Lysosomal/acidification genes (CTSK, ACP5)↑	[1,16,21]
CREB	GPCRs; c-Fms; RANK (indirect via cAMP)	cAMP↑; Ca^2+^	PKA–CREB; CaMK–CREB	Survival, early proliferation of precursors; cooperation with NFATc1/c-Fos	[1,16,19,20]
HIF-1α	O_2_ tension; integrins	ROS/NO	PHD inhibition → HIF-1α; PI3K–Akt/ERK crosstalk	Glycolytic shift; enhances resorptive capacity under hypoxia	[16,21]
FoxO (context-dependent)	Growth factor receptors; oxidative stress sensors	ROS (activates)	Akt–FoxO; JNK–FoxO	Limits excessive ROS; restrains over-resorption (homeostatic brake)	[1,16,20]
**Osteoblastogenesis**					
RUNX2	Integrins/FAK; Piezo1, TRPV4; LRP5/6–Frizzled (Wnt); BMP receptors	Ca^2+^ (SOCE via STIM1/ORAI1); ATP → P2Y; low ROS/NO; cAMP/PKA	Wnt/β-catenin; BMP–Smad1/5/8; ERK, p38; CaN–NFAT crosstalk	Lineage commitment; ALP↑; COL1A1↑; primes BGLAP/OCN	[1,2,10,11,12,13,25]
Osterix (SP7)	Integrins; BMP receptors; LRP5/6–Frizzled; Piezo1/TRPV4	Ca^2+^; ROS; cAMP	BMP–Smad1/5/8; Wnt/β-catenin; ERK1/2; AP-1 support	Maturation; COL1A1/1A2; SPP1/OPN; mineralization genes	[1,10,12,25,27]
ATF4	Integrins/FAK; PTH1R/GPCRs; amino-acid/ER-stress sensors	cAMP; Ca^2+^; ROS; metabolic cues	PKA–CREB/ATF4; PI3K–Akt–mTOR; PERK–eIF2α (ISR)	BGLAP/OCN↑; synthesis of collagen and other ECM components	[1,11,18,28]
NFAT (c1/c2 in osteoblasts)	Piezo1/TRPV4; GPCRs (P2Y); STIM1–ORAI1 (SOCE)	Ca^2+^ oscillations; ATP; NO; ROS → IP_3_R/SOCE)	PLC–IP_3_ → CaN–NFAT; crosstalk with ERK/p38	ALP, COL1A1; OPG↑	[1,2,10,11]
CREB	GPCRs (β-AR, PTH1R, P2Y); mechanosensitive tmACs	cAMP↑; Ca^2+^ (CaMKIV)	cAMP–PKA–CREB; Ca^2+^–CaMK–CREB; EPAC	Early proliferation/survival; primes matrix gene programs	[1,11,29]
BMPs/Smads; TGF-β/Smads; TGF-β/Non-smads	BMPR-I/II; TGFβR	ROS/NO fine-tune; Ca^2+^ minor role	BMP–Smad1/5/8 → Smad4; TGF-β–Smad2/3 → Smad4; non-Smad (TAK1–p38/JNK; PI3K–Akt)	BMP-Smads: RUNX2/SP7↑; osteogenic genes↑; TGF-β-Smads: COL1, CTGF, TIMP↑; OPG↑/RANKL↓	[1,12,25,30]
AP-1 (c-Fos/c-Jun)	Integrins/FAK; cytokine receptors; stretch-activated channels	ROS (↑MAPKs); Ca^2+^	ERK, JNK, p38 → AP-1	Proliferation; MMPs (remodeling); COL1 regulation	[1,2,18,21]
HIF-1α	O_2_ tension sensors; integrins; mechanosensitive receptors → NO production	ROS/NO	PHD inhibition → HIF-1α; crosstalk with PI3K–Akt/ERK	VEGF↑ (angiogenesis); glycolysis; osteogenesis–angiogenesis coupling	[1,22,31]

Abbreviations: NFATc1—nuclear factor of activated T cells, cytoplasmic 1; AP-1—activator protein-1; NF-κB—nuclear factor κB; MITF—microphthalmia-associated transcription factor; CREB—cAMP response element-binding protein; HIF-1α—hypoxia-inducible factor-1 alpha; FoxO—forkhead box O; RANK—receptor activator of nuclear factor κB; RANKL—RANK ligand; ITAM—immunoreceptor tyrosine-based activation motif; OSCAR—osteoclast-associated receptor; TREM2—triggering receptor expressed on myeloid cells 2; FcRγ—Fc receptor common gamma chain; αvβ3—integrin αvβ3; Piezo1—Piezo type mechanosensitive ion channel 1; TRPV4—transient receptor potential vanilloid 4; P2X7—P2X purinoceptor 7; P2Y—P2Y purinergic receptor; SOCE—store-operated Ca^2+^ entry; ATP—adenosine triphosphate; ROS—reactive oxygen species; NOX2—NADPH oxidase 2; NO—nitric oxide; TRAF6—TNF receptor-associated factor 6; MAPKs—mitogen-activated protein kinases; Syk—spleen tyrosine kinase; PLCγ2—phospholipase C gamma 2; CaN—calcineurin; OC—osteoclast(s); CTSK—cathepsin K; ACP5/TRAP—acid phosphatase 5, tartrate-resistant/tartrate-resistant acid phosphatase; TNFRs—tumor necrosis factor receptors; TLRs—toll-like receptors; M-CSF—macrophage colony-stimulating factor; CSF1R/c-Fms—colony-stimulating factor 1 receptor/c-Fms; ERK1/2—extracellular signal-regulated kinase 1/2; PI3K—phosphatidylinositol 3-kinase; Akt—protein kinase B; PKA—protein kinase A; CaMK—calcium/calmodulin-dependent protein kinase; CaMKIV—calcium/calmodulin-dependent protein kinase IV; PHD—prolyl hydroxylase domain enzyme(s); RUNX2—runt-related transcription factor 2; LRP5/6—low-density lipoprotein receptor-related protein 5/6; FZD—frizzled receptor; STIM1—stromal interaction molecule 1; ORAI1—calcium release-activated calcium channel protein 1; IP_3_—inositol 1,4,5-trisphosphate; IP_3_R—inositol 1,4,5-trisphosphate receptor; EPAC—exchange protein directly activated by cAMP; tmACs—transmembrane adenylyl cyclases; ALP—alkaline phosphatase; COL1A1—collagen type I alpha 1 chain; BGLAP/OCN—bone γ-carboxyglutamate protein/osteocalcin; SP7—osterix (Sp7 transcription factor); SPP1/OPN—secreted phosphoprotein 1/osteopontin; ATF4—activating transcription factor 4; GPCRs—G protein-coupled receptors; PTH1R—parathyroid hormone 1 receptor; ISR—integrated stress response; PERK—PKR-like endoplasmic reticulum kinase; eIF2α—eukaryotic initiation factor 2 alpha; β-AR—β-adrenergic receptor(s); BMPR-I/II—bone morphogenetic protein receptor type I/II; TGFβR—transforming growth factor-β receptor; Smad1/5/8—SMAD family member 1/5/8; Smad2/3—SMAD family member 2/3; Smad4—SMAD family member 4; TAK1—TGF-β-activated kinase 1 (MAP3K7); JNK—c-Jun N-terminal kinase; OPG—osteoprotegerin; VEGF—vascular endothelial growth factor; ER—endoplasmic reticulum; Wnt/β-catenin—Wnt/β-catenin signaling; BMP—bone morphogenetic protein; TGF-β—transforming growth factor-β. Legend: ↑ increase; ↓ decrease; → mark dirrection.

## 3. Photobiomodulation in Orthodontics: Focused Clinical Applications

PBM employs low-intensity red or near-infrared light to modulate biological responses without inducing thermal damage. PBM, which encompasses LLLT and LEDs, has gained increasing relevance in orthodontics. LLLT utilizes several light sources, including Gallium-Aluminum-Arsenide (GaAlAs) lasers (780–980 nm), Gallium-Arsenide (GaAs) lasers (∼904 nm, pulsed), Helium-Neon (HeNe) lasers (632.8 nm), and Diode lasers (660–980 nm). LEDs, in contrast, use non-coherent light. Owing to differences in wavelength, coherence, irradiance, spot size, and pulsing, lasers and LEDs exhibit distinct penetration profiles and dosing characteristics, making each suitable for specific dental applications [7,32].

Beyond orthodontics, PBM has been extensively studied for its analgesic, anti-inflammatory, and regenerative properties in contexts such as oral mucositis management, periodontal regeneration, and temporomandibular disorders [33,34]. Within orthodontics, PBM primarily modulates biological responses during tooth movement by enhancing osteoblast and fibroblast activity, promoting angiogenesis, and stimulating collagen synthesis. It also affects nociceptive pathways and inflammatory mediators [32,33,34,35,36]. Clinical studies consistently report that PBM reduces orthodontic pain, particularly within the first 72 h after force application, through the downregulation of proinflammatory mediators [37,38,39]. Preclinical evidence further suggests that PBM may attenuate orthodontically induced root resorption by suppressing osteoclastic activity in animal studies [40,41].

Additional applications include accelerating bone regeneration following corticotomy and rapid maxillary expansion [42]. PBM also appears to improve the stability and osseointegration of temporary anchorage devices (TADs) [43,44]. Furthermore, PBM supports periodontal and soft-tissue healing during OTM by stimulating collagen production and suppressing gingival inflammation, benefits that may be particularly relevant in patients with chronic periodontitis [37,45]. Table 2 summarizes light devices, their typical wavelengths, energy density/dosage, and main indications in the orthodontic practice.

## 4. Summary of Recent Systematic Reviews and Meta-Analyses on the Effects of PBM in Orthodontics

The primary objectives of utilizing photobiomodulation (PBM) in orthodontics are to accelerate tooth movement, reduce treatment duration, and alleviate pain. This review examined 20 systematic reviews and meta-analyses published between 2015 and 2023, which evaluated diverse outcomes, including overall acceleration, canine retraction, alignment and leveling, pain control, miniscrew stability, and safety concerns such as root resorption (Table 3).

Most reviews suggest that PBM accelerates OTM, although the observed gains are modest, and numerous trials report negligible effects. However, significant heterogeneity, particularly when considering pooled effects, and a substantial risk of bias limit the generalizability of these findings. In this context, it remains unclear whether the negative results stem from sub-optimal parameter choices or indicate a true lack of efficacy. Effects also vary according to the specific orthodontic procedure, such as canine retraction, incisor alignment, or intrusion [44,53,54,55,56,57,58,59,60]. In pediatric cohorts, reports indicate faster movement under PBM, although parameter choices are highly variable [47]. Broader reviews that encompass multiple orthodontic outcomes arrive at similar conclusions and also discuss potential periodontal benefits [61].

Pain-related outcomes appear to be more consistent, with many studies indicating that PBM effectively reduces early pain and chewing discomfort. Pain scores often decrease within the first 24 to 72 h following the placement of separators or archwires. However, certain findings lack consistency across studies and PBM protocols. High-quality systematic reviews suggest that the overall evidence remains weak, even when individual trials demonstrate the analgesic effects of the therapy [51,62,63,64].

Evidence regarding miniscrew stability is limited but suggests a possible benefit of PBM during the early healing phase [50,61,65,66]. With respect to safety, adverse events are rare and typically mild, consistent with PBM’s low-energy profile, non-ionizing light therapy. However, long-term data are sparse, and ongoing surveillance is warranted [55,57,61]. Findings on orthodontically induced inflammatory root resorption (OIIRR) remain inconclusive: while some studies suggest protective or reparative effects, others report potential exacerbation under specific conditions [52].

A recurring theme across reviews is the sensitivity of PBM outcomes to protocol parameters, including wavelength, power, fluence, session frequency, and cumulative dose. Reviews consistently highlight that the heterogeneity in these parameters accounts for much of the variation seen in pooled results and limits direct comparisons across trials [55,57,61,67]. Nonetheless, when photobiomodulation (PBM) is administered with appropriate dosing and scheduling, particularly during the first month of treatment, the likelihood of achieving clinically meaningful acceleration appears greater [53,57,58,59,67].

In summary, PBM represents a promising adjunctive modality in orthodontics, providing modest gains in acceleration and more reproducible short-term analgesia, although its effects on orthodontic interradicular root resorption (OIIRR) remain uncertain. To address current heterogeneity and mitigate risks of bias, robust, preregistered, sham-controlled randomized control trials (RCTs) with standardized dosimetry and standardized core outcomes are essential. Until such evidence is available, clinicians should apply PBM cautiously, ensuring optimization of dosing protocols, while closely monitoring both outcomes and safety.

**Table 3 biomedicines-13-02495-t003:** Systematic Reviews and Meta-Analyses on PBM/LLLT in Orthodontics.

References	Focus	n_studies	Key Result/Effect Size	Heterogeneity/Risk Bias	Bottom Line	Notes
Long et al. (2015) [53]	Acceleration (OTM)	5 RCTs	Subgroup: 780 nm & ~5 J/cm^2^ showed larger effects.	High bias	Weak evidence for small acceleration	Wavelength 780 nm; ~20 mW; ~5 J/cm^2^ subgroup showed benefits; GRADE weak
Imani et al. (2018) [54]	Acceleration (OTM) (canine retraction)	6 RCTs	Significant increase in rate of canine movement	Risk bias mixed (low, high, or undefined)	LLLT can accelerate canine distalization;	Various diode lasers (e.g., 808–980 nm); dosing heterogeneous
AlShahrani et al. (2019) [55]	Acceleration (OTM)	12 (RCTs + CCTs)	MD 0.59 favoring PBM; I^2^~95%	Low risk bias or undefined	Possible benefit, but heterogeneity limits certainty	Included GaAlAs & OrthoPulse; urged protocol standardization
Bakdach & Hadad (2020) [44]	Acceleration (OTM) (canine retraction)	25 RCTs	Statistically significant retraction acceleration reported	Variable protocols; risk bias is not clearly defined	Evidence graded low–very low; clinical significance uncertain	Recommends reporting total J/month rather than J/cm^2^
Jedliński et al. (2020) [56]	Acceleration (OTM) Canine retraction; incisors alignment; intrusion	6 RCTs; 8 meta analyses	PBM reduced treatment time	Risk bias is not clearly presented	Suggests benefit; protocol heterogeneity persists	Compared different lasers/parameters
Li et al. (2021) [57]	Acceleration (OTM)	8 RCTs + 1 quasi-RCT	Month 1 & 3 not significant; some benefit at month 2 in subgroups	Risk bias variable	Insufficient evidence overall; more RCTs needed	Split-mouth & parallel RCTs; varying wavelengths/energies
Yavagal et al. (2021) [47]	Acceleration (children)	14 (9 in meta-analysis)	Significant increase in movement	Risk bias heterogeneous	PBM optimal protocols unclear	Pediatric populations; diverse devices
Huang et al. (2023) [58]	Alignment-phase acceleration	8 (5 RCTs + 3 CCTs)	Significantly increased rate and reduced duration of alignment	Large; improved after subgrouping; risk bias heterogeneous	PBM accelerates alignment; protocol optimization needed	Included lasers and LEDs
Grajales et al. (2023) [59]	Acceleration (OTM)	18 RCTs (split-mouth only)	Trend toward faster movement; early (1 mo) effect not significant;	Present; Moderate risk bias	Evidence limited; more homogeneous trials required	Wavelengths ≤ 810 nm; ED ≤ 5.3 J/cm^2^ were associated with faster OTM.
Olmedo-Hernández et al. (2022) [60]	Acceleration (OTM)	22 (RCTs + CCTs)	No supportive evidence for LLLT effect on OTM in pooled analysis;	Present; high bias risk	No significant benefit; emphasized need for quality RCTs	Focused on acceleration; stringent inclusion; LLLT or LED
Ren et al. (2015) [62]	Pain after OTM	14 RCTs (659 pts)	LLLT reduced orthodontic pain by ~39% vs. placebo;	Present; High risk of bias in most RCTs	Analgesic effect probable; evidence quality low	Diode lasers; varied parameters
Shi et al. (2015) [63]	Pain (separator placement)	6 (5 RCTs and 1 CCTs)	Analgesic benefit noted; reliability limited	Present; variable risk/bias	Suggests reduced pain; more robust trials needed	Focus on separators
Deana et al. (2017) [64]	Pain during early OTM	20 RCTs	Reduced spontaneous & chewing pain at 24–72 h vs. placebo;	Present; high risk bias	LLLT likely reduces acute orthodontic pain; caution due to bias	Near-infrared LLLT subgroup
Li et al. (2015) [51]	Pain during OTM	11 RCTs	Pain reduction significant	Significant; different levels of risk bias	Insufficient evidence for routine use purely for pain relief	Called for higher-quality trials; LLLT and LED
Costa et al. (2021) [65]	Miniscrew (TAD) stability	6 (5 RCTs, 1 CCTs)	LLLT significantly increased OMI stability (e.g., Cohen’s d ~ 0.67)	Present; dominate low risk bias	Suggests stability benefit; more RCTs required	Early healing phases emphasized
Michelogiannakis et al. (2022) [66]	Miniscrew (MSI) stability	6 RCTs	2 RCTs positive; 1 no difference Increased stability in some subgroups after 2 mo	Low–moderate; insufficient data to analyze bias	Some evidence of improved stability; overall limited	LLLT parameters varied; called for more RCTs
Zheng et al. (2023) [50]	Miniscrew (MSI) stability	3 RCTs	Two RCTs ↑ stability; one no difference	Low–moderate; Risk bias was not clearly defined	Inconclusive but promising; more trials needed	Low-intensity lasers; split-mouth and parallel designs
Michelogiannakis et al. (2019 [52])	OIIRR during OTM	9 (mixed designs)	Conflicting—some reduction/reparative effect; others potential increase	Moderate–high; study designs mixed; risk bias is not clearly defined	Overall impact on OIIRR remains inconclusive	Human & animal data referenced
Domínguez Camacho et al. (2020) [67]	Effective wavelength range for acceleration	9 RCTs	Suggests 730–830 nm range as potentially effective; average speed improvement 24%	Variable heterogeneity and variable risk bias	Parameter guidance tentative; evidence base limited	Non-comparative synthesis; calls for standardization
Cronshaw et al. (2019) [61]	PBM across orthodontic applications	9 RCTs	Acceleration & post-op recovery (20–40% increase); evidence quality variable; better effects on pain reduction	High protocol variability	Encouraging but low–moderate certainty; protocol consensus needed	Overview (acceleration, pain, tissue healing); LLLT and LED

Abbreviations: DM-difference in mean; AMD-accumulative moved distance; RCTs-randomized control trials; CCTs-control clinical trials; MD-mean difference; ED-energy density; mo-month; OIIRR-orthodontically induced inflammatory root resorption. Legend: ↑ increase.

## 5. Molecular Aspects of Photobiomodulation and the Integration of Mechanical and Photon-Induced Signals During Orthodontic Tooth Movement

PBM utilizes red and near-infrared light to influence cell behavior through specific molecular pathways [7]. While these mechanisms are increasingly well understood across various tissues, direct evidence supporting PBM’s involvement in bone and periodontal remodeling during OTM remains limited. Nevertheless, several foundational findings can be applied to this context. This chapter aims to identify the initial cellular targets of photon energy and the earliest signaling triggers, briefly review the primary triggers of mechanical stress, and ultimately map their shared nodes, divergence points, and integration. The overarching goal is to delineate potential mechanisms through which PBM may modulate force-driven remodeling during OTM.

The primary photoreceptor for red and near-infrared light is cytochrome c oxidase (Complex IV), which is located in the mitochondria. When photons are absorbed, this process causes the dissociation of nitric oxide (NO) that is bound to the enzyme, leading to an increase in the mitochondrial membrane potential [68]. This alteration is linked to a temporary rise in adenosine triphosphate (ATP) synthesis and a regulated burst of reactive oxygen species (ROS) [7,69,70]. Furthermore, transient receptor potential (TRP) channels, located at the plasma membrane, constitute a second important class of targets for light. Their activation, particularly of TRPC1 and TRPV4, opens non-selective cation pores, leading to rapid Ca^2+^ entry [71]. In PDL cells and alveolar bone, this initial Ca^2+^ influx manifests as a sharp, early spike and represents a significant immediate response to photon exposure. Evidence supporting the link between TRP channels and PBM is seen in osteoblast-like Saos-2 cells, where pharmacological blockade of TRPV channels before PBM suppresses osteogenic outcomes [72]. TRPC1 is consistently associated with PBM-induced osteogenic and mesenchymal stromal/stem cell (MSC) responses, especially under 635-nm light. Additionally, PBM can activate TRPV1 in mast cells, resulting in increased Ca^2+^ influx [71,73].

In summary, ATP, NO, ROS, and Ca^2+^ act as essential signaling intermediates that regulate proliferation, differentiation, immune and inflammatory responses, and the remodeling of both hard and soft tissues [7].

Mechanical force is first detected at the membrane–ECM interface. This strain clusters β1-integrins and facilitates the formation of focal adhesions through the involvement of FAK/Src signaling and the subsequent opening of mechanosensitive calcium (Ca^2+^) channels. Two families of channels are especially important: Piezo1/2 and TRP channels, with TRPV4 and TRPC1 being notable members [74,75]. The production of various shared messengers, such as ATP, ROS, and NO, differs from the pathways stimulated by PBM. Mechanical deformation induces ATP release via pannexin-1 and connexin-43 hemichannels, and, in certain contexts, through Piezo-linked pores. Once released, extracellular ATP binds to purinergic receptors (P2X7/P2Y), leading to an increase in Ca^2+^ entry into cells [76]. ROS levels rise rapidly in response to mechanical stretch through the action of NADPH oxidases (NOX2/NOX4) and subsequently from mitochondria [77]. The production of NO is stimulated by load-sensitive isoforms of nitric oxide synthases (eNOS and iNOS). Mitochondria, while indirect targets during mechanical stress, play a crucial role by taking up Ca^2+^ through the mitochondrial calcium uniporter (MCU) [78]. Ca^2+^ serves as the key second messenger responsible for the transmission of signals triggered by both mechanical strain and PBM. However, an initial rapid spike in Ca^2+^ is not sufficient for effective signal propagation. The subsequent wave is essential for sustaining the response and involves the depletion of Ca^2+^ from the endoplasmic reticulum (ER) stores. This process is mediated by inositol 1,4,5-trisphosphate (IP3) and cyclic adenosine monophosphate (cAMP), which is generated from ATP by adenylyl cyclases [79]. As ER Ca^2+^ stores decrease, stromal interaction molecule-1 (STIM1) interacts with ORAI1 (calcium release-activated calcium modulator 1) channels to initiate store-operated calcium entry (SOCE). This process sustains a plateau of intracellular Ca^2+^ levels, keeping effectors active, and serves as a crucial mediator of the second wave of Ca^2+^ influx [15,80]. Figure 1A illustrates the proposed mechanisms involved in generating messengers during the combined effects of force and light.

Sustained Ca^2+^ activates several overlapping signaling pathways. The first is the Ca^2+^–calmodulin–calcineurin–NFAT pathway [81]. On the tension side, this pathway supports osteoblast-lineage programs, including RUNX2, ALP, and type I collagen. Additionally, it facilitates matrix deposition and mineralization [11,82]. In contrast, on the compression side, NFATc1 is involved in the differentiation of osteoclast precursors. The outcomes of these processes are influenced by the local RANKL/OPG ratio and the prevailing redox state [11,13,83]. Two in vitro models showed that LLLT at 808–810 nm upregulated NFATc1: (i) hPDLSCs under tensile strain [84] and (ii) RANKL-stimulated osteoclast precursors (RAW264.7 cells) [85].

Besides NFAT, sustained Ca^2+^ also drives the MAPK pathway. The MAPK cascade includes ERK1/2, c-Jun N-terminal kinase (JNK), p38, and ERK5 [86]. Through stepwise phosphorylation, these modules regulate transcription factor activity and are closely involved in both osteoclast resorption and osteoblast formation. Protein kinase C (PKC), co-activated by Ca^2+^ and diacylglycerol (DAG), provides a second entry into the MAPK network. ERK1/2 supports the proliferation of PDL fibroblasts and osteoblasts while upregulating the expression of early matrix genes. p38 promotes osteogenic differentiation and stress adaptation [87,88]. Both LLLT and LEDs stimulate MAPK pathways, particularly ERK1/2, in OTM models [69]. JNK activates activator protein-1 (AP-1), which can work with NFAT on composite promoters to activate both osteoblastic and osteoclastic genes [88].

PKC is also important for the activation of the NF-kB pathway. PKC is engaged when Ca^2+^ rises together with DAG. This process is followed by the translocation of the p65/p50 NF-κB dimer into the nucleus, where it activates target genes. In periodontal tissues, these target genes include COX-2, IL-1β, IL-6, TNF, MMPs, and RANKL [89]. The result is rapid matrix clearance on the compression side, along with the recruitment and maturation of osteoclasts, and regulated inflammation, while osteoblast activity is preserved on the tension side [69,71,90,91]. PBM can either reduce NF-κB–driven inflammation in PDL cells or promote NF-κB–linked osteoclastogenesis in precursor cells. In OTM, the overall effect depends on factors such as wavelength, fluence, timing (early vs. late phase), and the type of target cell [69,85].

The PI3K–Akt-mTOR axis integrates inputs from Ca^2+^, ATP, and growth factors. Akt promotes the survival and metabolic readiness of osteoblasts and PDL fibroblasts, reduces inappropriate apoptosis, and stabilizes β-catenin by inhibiting GSK3β [92,93]. mTOR complex 1 enhances protein synthesis and extracellular matrix production, and when dominant, it restrains autophagy [94,95]. Both ATP and ROS fine-tune nearly all Ca^2+^-driven signaling, as illustrated in Figure 1B. The role of NO differs slightly as it relates to both the stimulation and inhibition of cell signaling. The inhibition of NF-κB, Akt, and mTOR is mediated through S-nitrosylation mechanisms [96]. One paper demonstrated that LLLT (603 nm) stimulated rat MSC proliferation by stimulating the PI3K–Akt–mTOR pathway mechanisms [65,97].

Mechanical force and PBM share two additional signaling pathways. The first pathway is Wnt/β-catenin. Canonical Wnt signaling begins when Wnt ligands bind to Frizzled receptors and LRP5/6 co-receptors, which inhibits the β-catenin destruction complex. This inhibition allows β-catenin to translocate into the nucleus. Under mechanical load, the response mediated by β-catenin stimulates the commitment and maturation of osteoblasts on the tension side, while it restrains osteoclast activity and ECM mineralization on the compression side [98].

The second pathway involves TGF-β, which is activated by both mechanical force and PBM through distinct mechanisms. PBM directly activates latent TGF-β1 by inducing ROS-mediated oxidation of the latency-associated peptide (LAP), thereby releasing the active ligand. This activation favors Smad2/3 and Smad4 signaling, promoting the synthesis of the osteoblast extracellular matrix. Furthermore, an increase in OPG and a decrease in RANKL support pro-repair remodeling while simultaneously limiting inflammation. Mechanical stress activates latent TGF-β through the traction of αvβ6 and αvβ8 integrins on LAP–latent TGF-β-binding protein (LTBP) complexes, along with protease-dependent pathways at strained extracellular matrix sites. This activation results in spatially focused signaling. Non-Smad signaling pathways, which include TAK1, p38, JNK, and the PI3K–Akt–mTOR pathway, contribute to the subsequent alignment of the periodontal ligament (PDL) cytoskeleton, cell migration, and regulated inflammation. These signals also play a role in facilitating osteoclast formation on the compression side. In contrast, photobiomodulation (PBM) promotes a Smad-dominant, pro-repair signaling pathway. It is therefore intriguing to consider how these differing outcomes are coordinated during the simultaneous application of light and mechanical force [69,99,100,101,102].

An excellent review by Wang et al. [98] shows that the beneficial effects of PBM on OTM primarily engage NF-κB, MAPK (ERK), Akt, and BMP/Smad signaling. Thus, our hypothesis about the involvement of additional PBM-transmitted signals in force-mediated pathways is extrapolated from other non-OTM models.

Figure 1B illustrates signaling generation during combined mechanical loading and PBM. Complementing the text, the figure presents a broader view of integration and highlights the main points of convergence. However, the underlying network is even more complex and extends beyond the scope of this review.

In summary, PBM overlays mechanical-force signaling by delivering the same messengers, Ca^2+^, ATP, low-level ROS, and NO, via TRP channels and mitochondrial cytochrome c oxidase. These signals most probably converge on shared hubs such as Ca^2^-calcineurin–NFAT, MAPK (ERK, p38, JNK), PI3K–Akt–mTOR, NF-kB, TGF-β/Smad, BMP/Smad, and Wnt/β-catenin. We hypothesize that PBM fine-tunes force-driven mechanotransduction in a phase- and site-dependent manner. It may amplify or modify early signals when acceleration is desirable and stabilize reparative pathways during consolidation.

## 6. Photobiomodulation and Cellular Crosstalk in Orthodontic Tooth Movement: Cellular Dynamics, Matrix Remodeling, and Vascular Adaptation

During OTM, the interaction between various stromal components and infiltrating cells in the PDL and alveolar bone is key to determining the outcome of OTM. A well-recognized aspect of this process is the bidirectional communication between osteoblasts and osteoclasts. These interactions are mediated through membrane receptor-ligand connections and soluble factors [103].

Osteoblasts secrete several key factors that regulate osteoclast development, including RANKL, M-CSF, and WNT5A. M-CSF and RANKL work synergistically to promote the differentiation, fusion, and activation of osteoclasts. In addition to these cells, other types contribute to this complex network. Osteocytes produce RANKL, SOST, and PGE2. PDLSCs also secrete RANKL, PGE2, and IL-6. SOST stimulates the formation of osteoclasts by inducing RANKL while simultaneously inhibiting Wnt signaling through the LRP5/6–Frizzled receptor complex, thus suppressing osteoblast differentiation. PGE2 enhances the differentiation and function of osteoclasts on the compression side by inducing RANKL expression in PDLSCs, osteoblasts, and osteocytes, primarily via the EP4 receptor [16,74,103,104]. In contrast, Semaphorin 3A (SEMA3A), expressed on osteoblasts, binds to neuropilin 1 (NRP1) on osteoclasts and inhibits the RANKL-mediated formation of osteoclasts [105].

As previously mentioned, RANKL binds to RANK receptors on monocyte-derived precursors, facilitating their fusion into multinucleated osteoclasts [106,107]. RANKL is primarily produced by osteoblasts and PDLSCs within the PDL and osteocytes [16,74]. Osteoclasts are key cells that coordinate osteolysis and the breakdown of the ECM on the compression side during the initial phase of OTM. Consequently, it is hypothesized that PBM, by interacting with osteoclasts, may influence osteoclast-mediated processes. This hypothesis is supported by several studies. LLLT has been shown to stimulate osteoclastogenesis on the compression side by upregulating RANKL and osteoclast markers [108,109,110]. Various in vivo rat models of OTM demonstrated an increase in the velocity of tooth movement, which correlated with the upregulation of RANKL following LLLT, as well as an increase in M-CSF, RANK, MMP-9, cathepsin K, and MMP-13 [111]. A split-mouth human study indicated that LLLT (810 nm, 100 mW; 6.29 J/cm^2^ applied on days 0, 7, 14, and 21) accelerated tooth movement [110]. This acceleration was associated with a significant increase in RANKL levels and a decrease in OPG levels in gingival crevicular fluid (GCF) on the irradiated side over the first 28 days of canine retraction. Similar clinical outcomes were confirmed in another study involving human participants, although different laser parameters (670 nm, 200 mW, and 6.37 W/cm^2^) and treatment protocols were utilized [112].

LLLT appears to also target other molecules that are important for the interaction with osteoclasts. Regarding SOST, most studies in OTM and non-OTM bone models revealed that LLLT suppresses SOST expression and enhances osteoblastogenesis. These findings underscore the role of SOST in osteoclastogenesis and suggest that targeting this molecule with LLLT is particularly relevant for promoting osteoblast activity on the tension side during OTM [74,113]. The findings on PGE-2 vary. One study reported an increased level of PGE-2 in GCF in a human split-mouth RCT involving 810 nm GaAlAs LLLT at days 7 and 21 compared to the non-irradiated side. This increase in PGE-2 (and IL-1β) in GCF at those time points suggests a potential role in accelerating OTM due to its pro-osteoclastogenic effects [114]. However, another human double-blind clinical trial with a quadrant design found that diode laser therapy applied on days 0, 2, 18, and 30 during retraction resulted in a progressive reduction in PGE-2 in GCF compared to the control group. The authors interpreted these results as indicative of analgesic and anti-inflammatory effects [115].

During OTM, remodeling of the ECM is crucial because it facilitates the migration of osteoclasts, fibroblasts, endothelial cells, and immune cells through the PDL microenvironment. These processes are also bidirectional. The remodeling is dominantly caused by the action of MMPs, a group of zinc-dependent enzymes that break down ECM components. It is known that mechanical stimuli rapidly increase the levels of MMPs in the area of compression. Fibroblasts, osteoclasts, and endothelial cells in the PDL dominantly produce these enzymes. MMP-1 and MMP-8 degrade fibrillar collagens, whereas MMP-2 and MMP-9 break down denatured collagens, gelatin, and elastin [116]. Tissue inhibitors of metalloproteinases (TIMPs), especially TIMP-1 and TIMP-2, tightly control the activity of MMPs. Signaling pathways such as MAPK and NF-κB, involved in mechanotransduction, control their production and activity. The MMP/TIMP ratio controls ECM breakdown and is balanced by the MMP/TIMP ratio. For example, the compressive stress augments MMP-9 levels, temporarily decreases TIMP levels, and thus increases the MMP/TIMP ratio. This is followed by an increase in ECM breakdown [117].

Several clinical and experimental studies indicate that PBM modulates the catabolic phase of OTM by increasing MMP expression. For example, Jivrajani and Bad Patil (2020) demonstrated a significant rise in MMP9 concentrations in GCF during the initial phase of OTM [118]. Lee et al. [119] showed that LLLT significantly increased the relapse rate in a rat model of OTM, which positively correlated with the expression of several MMPs (1, 2, 8, 9, and 13) i the PDL. The LLLT did not significantly change the relapse rate and MMPs expression in the group treated with doxycycline (a potent TIMPs inhibitor), compared to the corresponding control [119]. Mechanical force increases the expression of intercellular adhesion molecule-1 (ICAM-1) and vascular cell adhesion molecule-1 (VCAM-1). These molecules are important for the adhesion and transmigration of leucocytes from the blood to the PDL. NO, which is an important mediator of vasodilation and a regulator of osteoclasts and immune cell functions, is produced by the increased activity of eNOS [120,121]. Although modest, these results support the hypothesis that PBM could increase or modulate the remodeling during the early phase of OTM.

Osteoclasts are a potent modulator of osteoblasts. For example, the binding of ephrin B2 (EFNB2) on osteoclasts with EPHB4 on osteoblasts promotes the differentiation of osteoblasts [103,122]. Osteoclasts secrete signaling molecules such as sphingosine-1-phosphate (S1P), collagen triple helix repeat containing 1 (CTHRC1), and complement component C3, all of which enhance osteoblast differentiation. Additionally, TGF-β and insulin-like growth factor 1 (IGF-1), released from the bone matrix during osteoclastic bone resorption, stimulate osteoblast-mediated bone formation [16].

Numerous studies, performed dominantly in vitro on PDL components, showed the direct stimulatory effect of PBM, dominantly using in vitro cell models that can be translated to OTM. By using a Nd:YAG laser (1064 nm) of different energy output to treat hPDLDCs, Wang et al. (2022) showed that LLLT significantly stimulated cell proliferation and osteogenic differentiation through BMP/Smad signaling [123]. Similar results with the same laser type have been recently confirmed on hPDLSCs, and the authors identified the involvement of the stromal-derived factor-1 (SDF-1)/CXCR4 signaling pathway in these processes [124]. Another study demonstrated that PBM therapy with a 940 nm diode laser (4 J/cm^2^) increases VEGF and BMP expression in hPDLSCs, along with enhanced mineralization. However, the proliferation, as determined with the MTT test, was not changed [125]. LLLT and LEDs both increased type I collagen and osteonectin in human osteoblasts, but only LLLT activated proliferation through the ERK1/2 pathways, indicating that they modulate osteoblast function via distinct mechanisms [126].

Based on a systematic review of in vitro studies, Mylona et al. [127] reported that PBM with lasers within the 630–830 nm wavelength range can enhance both the stemness and differentiation potential of PDLSCs. Regarding fluence, it was suggested that doses should not exceed 4 J/cm^2^ [127]. Zang et al. [128] showed that LLLT applied under tension stress stimulated PDL cell proliferation and promoted periodontal remodeling. It suppressed osteogenic markers while increasing osteoclast-related factors, collagen, MMPs, and TIMPs. The authors hypothesized that LLLT may more strongly stimulate TIMPs than MMPs, potentially limiting collagen degradation and enabling net collagen accumulation. However, the dynamic balance of MMPs/TIMPs during OTM remains poorly defined and warrants further investigation. The increased expression of TIMPs during the anabolic phase is crucial for inhibiting MMPs and promoting the synthesis of new fibers and ECM components.

Except for the role in ECM turnover during OTM, LLLT enhances the formation of the new capillary network in periodontal tissues on the tension site upon application of mechanical force by stimulating the expression of VEGF. The signaling pathways involved are those transmitted via HIF-1α and ERK activation [129,130]. These responses are most evident in the areas of active remodeling and are essential for maintaining tissue viability during OTM. Importantly, VEGF stimulates both angiogenesis and MMP expression. However, it is not known how these opposite processes are directed on the compression versus tension side. Vascular and ECM remodeling processes are closely linked. VEGF stimulates both angiogenesis and MMP expression, and both mechanisms, together with NO, are important to facilitate osteoclast recruitment and ECM degradation [131,132].

The effects of PBM on endothelial functions and vascular remodeling during OTM have not been sufficiently examined. Zhong et al. [133] showed that PBM (980 nm wavelength; diode laser) significantly increased the formation of blood vessels and new bone generation in a rat OTM model. In a co-culture model of human umbilical vein endothelial cells (HUVEC) and osteoblast precursor (MC3T3-E1 cells), the same authors showed an increase in HUVEC proliferation and up-regulation of angiogenic-related genes (HIF-1α and VEGF) [133]. Another study also demonstrated that LLLT stimulates the proliferation, migration, and angiogenic functions of HUVEC. Several signaling molecules and pathways are involved in these mechanisms, including HIF-1α, eNOS, VEGF, and PI3K/Akt/mTOR [134]. A systematic review by Berni et al. [135] presents in vitro and in vivo models of bone regeneration. The authors concluded that LLLT facilitates angiogenesis, supports fracture repair, and induces osteogenic differentiation of MSCs.

Overall, the effects of PBM on bone repair and remodeling largely mirror the findings from OTM models. Rather than being generalized, these effects are validated in other LLLT applications. In this context, a study performed by Szymczyszyn et al. (2016) indicates that the transdermal application of LLLT improves endothelial function in the skin by enhancing antioxidant and angiogenic responses [136].

Table 4 shows the simplified mechanisms of the integration of PBM- and OTM-generated signals at the levels of cellular and extracellular crosstalks.

In conclusion, this review supports the view that PBM can modulate the crosstalk among major cellular players in OTM and the signaling that drives ECM and vascular remodeling. However, the current evidence is modest, and many gaps remain. Therefore, further advances are needed to enable more reliable translation into orthodontic practice. From an orthodontic perspective, it is necessary to better standardize force-system parameters, the temporal profile of force delivery, appliance characteristics and protocols, anchorage strategy, instrumentation and calibration, and clinical outcomes within the biological context. In parallel, standardization of light parameters, such as wavelength (nm), power (mW) and irradiance (mW·cm^−2^), fluence (J·cm^−2^) and exposure time, as well as emission mode (continuous or pulsed with frequency/duty cycle), is of particular significance. Moreover, the choice of appropriate spot size and anatomical placement (tension/compression sites), session frequency and cumulative dose, timing relative to activations/reactivations, and any co-interventions is an urgent need.

## 7. Apoptosis and Autophagy in Orthodontic Tooth Movement: Mechanosensitive Responses and Cellular Interplay Under Photobiomodulation

Apoptosis and autophagy are two interconnected processes related to cell self-destruction, but often with opposite directions during OTM. They regulate the dynamics of osteocytes, osteoblasts, PDLSCs, and immune cells turnover. Apoptosis removes damaged or overstimulated cells. Autophagy supports survival by clearing damaged organelles, limits ROS production, and tunes inflammatory signals. It is supposed that their counterbalance helps the coordination of bone resorption and formation during OTM [137].

Autophagy is a key cellular response to mechanical stress during OTM. In this context, Li et al. [138] provide a comprehensive review showing that mechanical force influences autophagy differently depending on the stressed side (compression vs. tension) and the target cell type. On the compression side, autophagy is generally stimulated both in stromal and infiltrating cells. However, the outcomes vary depending on the target cells. The compression stress increases autophagy in PDLSCs, osteoblasts, and osteocytes, but decreases it in cementoblasts. In osteoclasts, autophagy can be either down- or upregulated, depending on the experimental setup. On the tension side, mechanical force induces autophagy in all cell participants, accompanied by enhanced osteogenesis and cementogenesis, while reducing cell death and osteoclastogenesis. These patterns are supported by multiple in vivo and in vitro studies [139,140,141].

Hypoxia plays a crucial role in inducing autophagy, which helps cell survival during nutrient deprivation. As previously mentioned in this review, compression creates hypoxic conditions. Hypoxia activates autophagy in nearly all cells involved in OTM through the action of HIF-1α. On the compression side, it promotes osteoclastogenesis but inhibits osteoblastogenesis. Experimental evidence supports this conclusion, demonstrating that hypoxia-induced osteoclastogenesis can be diminished by treating osteoclast precursors with an autophagy inhibitor (3-MA) or by knocking down ATG5. Furthermore, autophagy has been shown to reduce hypoxia-induced apoptosis in cementoblasts [138]. Given that OTM involves the application of mechanical force in hypoxic conditions, it is essential to design experiments that simultaneously model these processes. The outcomes of such experiments may be additive or synergistic, but they could also yield unpredictable results.

Mitophagy is a specialized form of autophagy that selectively eliminates damaged mitochondria, thereby maintaining mitochondrial quality and ensuring cellular homeostasis [142]. In vitro studies have shown that mitophagy is activated in PDLSCs when subjected to both compressive and tensile forces, which positively affects OTM. Furthermore, mitophagy facilitates osteoblastic differentiation in PDLSCs. This conclusion is reinforced by experiments indicating that Urolithin A, a known mitophagy inducer, enhances bone remodeling and accelerates OTM, while the mitophagy inhibitor Mdivi-1 induces opposing effects [143,144].

LLLT is a recognized modulator of autophagy. In MC3T3-E1 pre-osteoblasts exposed to H_2_O_2_, 808 nm LLLT enhanced autophagic flux, indicated by an increase in LC3B and a decrease in p62 through the inhibition of the PI3K/Akt/mTOR pathway. This process subsequently promotes osteogenic differentiation and matrix formation. Blocking autophagy reverses these effects [145]. In osteoporotic bone marrow-derived stem cells, 650 nm PBM activated autophagy, leading to increased ALP activity and enhanced mineral deposition [146]. In a muscle degeneration model, Fernandes et al.demonstrated that PBM upregulated the expression of autophagy-specific molecules, including SQSTM1/p62, Beclin, and Parkin [147]. This activation is followed by the stimulation of AMPK and the initiation of intercellular signaling mediated by TGF-β. 

The integration of these limited PBM findings into the autophagy pathways in OTM remains unclear. Given that autophagy plays both protective and regulatory roles in OTM, selecting appropriate LLLT parameters could potentially enhance OTM by promoting autophagy. In this regard, a foundational understanding of the key regulatory points of autophagy may be beneficial. We believe that influencing mitophagy through PBM may be more relevant than modulating classical autophagy, as mitochondria are the primary targets of PBM. Numerous studies on muscle and neurodegenerative models indicate that PBM serves as a significant modulator of mitophagy. It has the potential to suppress excessive injury-induced mitophagy during acute tissue damage and to correct impaired mitophagy in chronic degenerative processes [148,149,150].

Several studies show the significance of apoptosis in OTM. Mechanical loading can trigger apoptosis in several periodontal cell types, particularly in PDLSCs and osteocytes. Osteocytes are the first cells responding by both apoptosis [151,152] and necroptosis at the early OTM phase. Necroptosis, as a special form of cell death, is influenced by TNF produced by PDLSCs and inflammatory cells. The released DAMPs from dead osteocytes are important stimulators of macrophages and osteoclasts [153]. In mechanically stressed cells, autophagy precedes and often transitions to apoptosis once hypoxia and nutrient deprivation prevent maintenance of vital functions. If the force is pronounced, apoptosis may be triggered directly, bypassing autophagy.

In a mouse OTM model, TUNEL staining showed an increase in osteocyte apoptosis on the compression side by 12 h, with a peak at 24 h [154]. PDLSCs have been demonstrated to be an appropriate model for investigating the impact of mechanical force on apoptosis [155]. Wang et al. (2021) showed an increase in expression levels of caspase-3 (a hallmark of irreversible apoptosis) as early as 1 h of OTM induction, and the process was gradually enhanced following the 1st day [139]. Similarly, a study on human periodontium extracted after OTM showed an increase in the apoptotic index from day 3 on both compression and tension sides. Apoptosis was maximal at day 7, followed by a gradual decrease by day 21 [156].

In vitro experiments support these findings. It has been shown that cyclic or static mechanical stress activates caspase-3 and caspase-5 in PDLSCs and MG-63 osteoblast-like cells. The apoptotic process was force-dependent [157]. Yu et al. [158] further confirmed similar findings in MG-63 cells subjected to 15% cyclic stretch at 0.1 Hz based on increased cleaved poly (ADP-ribose) polymerase (cPARP) and caspase-3 expression. Interestingly, periostin exerted a protective, anti-apoptotic effect during the application of mechanical force [158]. Li et al. (2021) reviewed several in vitro studies on PDLCs and MG-63 cells and concluded that mechanical loading consistently triggers apoptosis, primarily through caspase activation [159].

Recent findings highlight the functional relevance of apoptosis in bone regeneration. Liu et al. demonstrated that apoptosis of PDLSCs activates Lepr^+^ osteoprogenitors, particularly on the compression side of the PDL. These cells are a special subpopulation of osteoblast precursors. PDLSC-derived apoptotic vesicles can directly drive osteoblastic differentiation of Lepr^+^ cells, thus linking force-induced apoptosis to osteogenesis in OTM [160]. Shen et al. [161] showed that inhibition of Piezo1 reduced force-induced apoptosis in periodontal tissues and fibroblasts via downregulation of the p38/ERK1/2 pathway. These findings indicate that modulation of mechanotransduction signals can influence cell survival under orthodontic load [161]. However, some studies suggest that PDLSCs may adapt to compressive stress without undergoing significant apoptosis [162]. The apoptotic rate in OTM is directly proportional to force intensity. Stronger or higher-amplitude forces shift signaling toward apoptosis, and this is particularly relevant for the pathogenesis of OIIRR. Contrary, lighter forces favor cell survival [163,164].

Although no studies have directly assessed the effects of PBM on apoptosis in OTM models, evidence from other systems offers important insight. In general, PBM inhibits apoptosis in many cells and thus promotes cell survival. Zhang et al. (2010) demonstrated that this is achieved through selective activation of the PI3K/Akt pathway and suppression of the GSK3β/Bax pathway [165]. Generally, LLLT exhibits biphasic effects depending on energy dose and cell type. At lower doses, photon energy can protect cells, such as endothelial or muscle cells, from apoptosis during inflammatory or oxidative stress conditions. In contrast, higher doses can activate apoptotic pathways and promote cell death. [166]. This dose-dependent duality underscores the complexity of LLLT-induced cellular responses [167,168,169].

From an orthodontic perspective, it is essential to choose light parameters that promote the catabolic phase of OTM and prevent premature apoptosis so that osteoclast function is maintained. Nevertheless, once pro-apoptotic mechanisms are initiated, they should not be suppressed, as their timely progression helps trigger subsequent anabolic processes. Therefore, using low laser energy and a smaller irradiation field may be advantageous in the initial phase to suppress apoptosis and promote autophagy at the same time. As OTM progresses into the transition and consolidation phases, it is reasonable to consider the cytoprotective effects of PBM. This approach may help preserve cementum, the periodontal ligament (PDL), and osteocytes, while also supporting fiber reorganization and osteogenesis on the tension side. Such an approach can be particularly beneficial for patients with thin roots, previous external apical root resorption (EARR), or when OTM occurs in an inflammatory environment, such as periodontitis. In these instances, a localized application with lower fluence and a moderate treatment schedule could be recommended. However, these hypotheses require validation through targeted animal and clinical studies.

## 8. Immune Dynamics in Orthodontic Tooth Movement: Unresolved Mechanisms of Photobiomodulation

### 8.1. Innate Immunity

Innate immunity plays a crucial role in OTM. Neutrophils and monocytes are key components of the innate immune response, making up approximately 70% of leukocytes in the PDL of mice [89]. Neutrophils are the first responders to mechanical force. They rapidly infiltrate the PDL, participate in phagocytosis, degrade the ECM components, recruit monocytes and macrophages, and subsequently undergo apoptosis [89,170,171]. During this process, they release various cytokines and chemokines, including IL-1, IL-6, TNF-α, CCL2, CXCL1α, and CCL3, which facilitate osteoclastogenesis and bone resorption [172]. In humans, the number of neutrophils in GCF and saliva increases within 2 h after the application of force [173].

Several studies have indicated that PBM can influence neutrophil activity. LLLT has been shown to enhance oxidative burst and fungicidal activity in human neutrophils [174]. Additionally, irradiated peripheral blood granulocytes exhibit higher fMLP-induced chemiluminescence at 632.8 nm and 0.6 J/cm^2^ [175]. PBM can also elevate levels of IL-1β, CCL2, and CCL3 in peripheral blood mononuclear cells (PBMCs) in a dose-dependent manner [176].

Monocytes serve two primary functions. They clear apoptotic cells and act as precursors for osteoclasts, macrophages, and dendritic cells (DCs) [171,177]. Their infiltration typically peaks around day 3 following the application of force. Subsequently, osteoclast differentiation occurs on the compression side, contributing to bone resorption [177]. The depletion of monocytes and macrophages has been shown to reduce osteoclast formation, lower levels of TNF and iNOS, and significantly impede tooth movement in mice [138,178]

Macrophages control inflammation and remodeling. On the compression side, M1 macrophages play a role in breaking down the matrix, removing debris, and activating osteoclasts [138,172,173]. The number of macrophages goes up on day 3, reaches the highest point on day 7, and remains high until day 14 [89,171,178]. Around day 7, the cells upregulate chemokine signaling and transcribe more inflammatory genes [179]. At baseline and during OTM, CCR2^+^ macrophages are the most common type of macrophage. Blocking CCR2 slows tooth movement and lowers inflammatory signals [179]. In vitro, mechanically activated macrophages secrete proinflammatory cytokines that stimulate PDL cells and facilitate osteoclastogenesis [180].

M2 macrophages, which are differentiated from monocytes or M1 macrophages inhibit inflammation and support healing and bone production by increasing BMP2. At the same time, they suppress osteoclastogenesis [181]. Moreover, M2 macrophages promoted osteogenic differentiation in PDLSCs via IL-10 and reduced the RANKL/OPG ratio [182]. In rat OTM models, the frequency of M2 macrophages increases during the later stages, coinciding with the onset of bone and PDL modeling and a reduction in inflammation [183]. In vitro studies also support this phenomenon. In human osteoblast co-cultures, M2 macrophages promoted the proliferation of osteoblasts, in contrast to M1 macrophages, which had an opposing effect [89,171]. In summary, M2 macrophages, along with their EVs, enhance the proliferation, differentiation, and mineralization of osteoblasts through various signaling pathways, either directly or indirectly via PDLSCs. Additionally, these cells promote angiogenesis through VEGF and contribute to ECM modeling by modulating the MMP/TIMP ratio [180,181,184].

PBM can influence macrophage responses. At a wavelength of 660 nm, LLLT elevated M1-associated cytokines and chemokines in monocytes and macrophages, and it was associated with histone modifications at the TNF-α and IP-10 loci [185]. In RAW264.7 cells, PBM increased M1 markers in naïve M0 cells. However, when the cells were prepolarized toward the M1 type, PBM reduced M1 markers and increased M2 markers through the PI3K/AKT/mTOR signaling pathway [186]. These findings suggest that PBM initially stimulates M1 priming, followed by a transition to the M2 phenotype and resolution. Fernandes et al. noted an increase in IL-6 levels at 660 nm in M1 cells, although a reduction in other pro-inflammatory cytokines was observed [187] Zhang et al. (2019) demonstrated that 810 nm light reduced M1 markers, enhanced M2 polarization, and increased the production of neurotrophic factors through PKA/CREB signaling in macrophages derived from mouse bone marrow [188]. PBM-stimulated M2 macrophages also secrete PDGF, which may promote osteogenesis and matrix synthesis via its receptor and the PI3K–Akt/ERK signaling pathway [186,189,190]. Table 5 outlines the distinct ways in which M1 and M2 macrophages regulate catabolic and anabolic processes under the combined influence of photon light and mechanical force. Additionally, various other signaling pathways are presented. The overlapping effects of PBM are suggested primarily based on its impact observed in models outside the OTM context.

The role of other innate immune cells in OTM, such as DCs, gamma delta (γδ) T cells, and natural killer (NK) cells, remains underexplored, and there is limited understanding of how PBM might affect their functions [171,172]. Innate lymphoid cells (ILCs) have only recently been identified in the context of OTM. These cells respond rapidly and do not require antigen specificity. For example, ILC1s migrate toward force-stressed periodontal ligament (PDL) cells, whereas ILC2s and ILC3s proliferate in co-culture settings [196].

Based on existing information, we propose that PBM can assist in the early stages of OTM by enhancing pro-inflammatory signaling. Clinical investigations have shown elevated IL-1β, IL-6, IL-8, and RANKL in GCF during laser-assisted OTM [110, 191–193]. Additionally, a systematic study reported increases in IL-1β, IL-8, OPN, and PGE2 in GCF, while TNF-α and TGF-β showed no consistent changes. The levels of IL-6, RANKL, and OPG varied based on timing and dose [194]. Mechanistically, PBM may directly influence innate immune cells, providing a transient early boost. However, the evidence remains inconclusive, as other inflammatory models frequently demonstrate suppressed functions of neutrophils and monocytes in conditions like periodontitis and RA [195,197]. Overall, the situation appears heterogeneous and likely biphasic. In the early phase, PBM could enhance the functions of neutrophils, monocytes, and macrophages, as well as RANKL-linked osteoclastogenesis, serving as an additive mechanism in supporting inflammation. When early inflammation evolves into the chronic phase, PBM may suppress overactive inflammatory responses, thereby preventing orthodontically induced root resorption (OIRR). This aligns with most data indicating that PBM primarily exerts an anti-inflammatory effect on PDLSCs. In the later phase, PBM may assist in consolidation by promoting M2-mediated resolution, increasing angiogenesis, and stabilizing the extracellular matrix [198,199].

Figure 2 provides a simplified illustration of the role of innate immune cells in PBM-modulated OTM.

### 8.2. Adaptive Immunity

Compared with innate immunity, adaptive immunity remains insufficiently studied in OTM. Several investigations showed that T cells infiltrate the PDL early during OTM, and the process is guided by pro-inflammatory mediators released from PDL cells, neutrophils, and M1 macrophages [2,200]. Rodent models underscore adaptive immunity as a crucial regulator of tissue remodeling. Th1 and Th17 cells predominate in the initial phase, in comparison with other CD4^+^ T helper subsets.

It has been shown that Th1 cells express RANKL and facilitate osteoclastogenesis, often through the indirect regulation of macrophages and dendritic cells (DCs) via IFN-γ. IFN-γ, produced by Th1 cells, cytotoxic T cells, DCs, and natural killer (NK) cells, is a well-known potent activator of M1 macrophages. However, some experimental data suggest that it may also reduce osteoclast formation and, consequently, tooth mobility, though the net effect can vary depending on the specific context. For instance, IFN-γ accumulation is observed alongside increased trabecular volume and reduced trabecular spacing at compression sites, indicating a potential bone-protective role [201,202]. However, the role of IFN-γ in OTM remains intricate.

Th17 cells secrete IL-17A, IL-17F, and IL-22, the cytokines that promote inflammation and drive osteoclast formation [203]. Th17 cells support RANKL-mediated osteoclastogenesis more strongly than other T-cell subsets [204]. IL-17 activates fibroblasts, PDLSCs, and epithelial cells to produce chemokines (e.g., CCL2, CXCL8) and MMPs, facilitating ECM remodeling and the recruitment of immune cells [205]. It also stimulates the production of IL-6, IL-8, and PGE2 from epithelial, endothelial, and stromal cells. At the same time, IL-17 augments osteoblast RANKL expression, so promoting periodontal bone resorption [206]. Increased IL-17 levels in GCF may signify root resorption during OTM [207].

The role of Th2 cells in OTM is not well established. IL-4, a pivotal Th2 cytokine, impedes RANKL- and TNF-α–induced osteoclastogenesis by inhibiting NF-κB and MAPK pathways, particularly the NFATc1 activation through STAT6 and NF-kB-mediated signaling programs. At the same time, they promote osteoblastogenesis. IL-4 also indirectly reduces pro-inflammatory mediators such as TNF-α, IL-1, and IL-6 [208,209].

The role of cytotoxic (CD8+) T cells is less explored compared to that of CD4+ T cells. CD8+ T cells secrete RANKL, TNF-α, IL-1, IL-6, and IL-17, which promote osteoclast formation. In contrast, these cells also activate Wnt signaling to stimulate osteoblasts. Additionally, FoxP3+ CD8+ T cells inhibit osteoclastogenesis through mechanisms mediated by IFN-γ and CTLA-4 [210,211].

During the initial and linear phases of OTM, there is an increase in the number of CD220 B cells. Although these lymphocytes are a source of RANKL, their specific role in OTM remains uncertain. Only one study has suggested that hPDLSCs may suppress B-cell activity through direct contact [212,213]. It is supposed that B cells may have dual functions in OTM. They may promote osteoclastogenesis by secreting IL-6, GM-CSF, and RANKL, while also indirectly stimulating bone resorption via IL-12–mediated Th1 activation. At the same time, they produce anti-inflammatory mediators such as OPG, IL-10, and TGF-β, which inhibit RANKL and osteoclast differentiation [214,215]. Despite these regulatory functions attributed to B regulatory cells (Bregs), direct evidence of B cells functioning as significant modulators in OTM is limited.

In contrast to innate immunity, the effects of PBM on adaptive immunity during OTM remain poorly characterized, and most available insights come from non-skeletal models. Some direct evidences come from in vitro studies showing that LLLT enhances T-cell proliferation stimulated by phytohemagglutinin (PHA) and B-cell proliferation in the presence of *Staphylococcus aureus* [216,217,218]. However, other findings are inconsistent. For example, Al Musawi et al. reported that 589 nm irradiation of whole blood (72 J/cm^2^) increased the total number of CD45^+^ lymphocytes and NK cells but did not impact T- or B-cell subsets [219]. Additional in vitro and in vivo models suggest the T-cell stimulatory effect of PBM. In a murine neuroinflammation model, PBM activated the JAK2/STAT4/STAT5 pathway and promoted the release of IFN-γ and IL-10 from CD4^+^ T cells [220]. In tumor models, fractional CO_2_ laser treatment resulted in an increase in local CD4^+^ and CD8^+^ T-cell subsets [221]. However, periodontitis models, while not directly investigating T cell phenotypes, consistently demonstrate the anti-inflammatory effects of PBM, including pathways related to Th1 and Th17 cell immunity [222,223].

During the later stage of OTM, a transformation in the composition of Th cell subsets occurs, marked by an increase in Tregs [35,199,202]. Tregs, a subpopulation of CD4^+^ T cells distinct from other Th subsets, play a crucial role in preserving immunological homeostasis and regulating excessive inflammatory responses. These effects arise through direct cell–cell interactions and through the secretion of anti-inflammatory cytokines, notably IL-10 and TGF-β [224]. During OTM, Tregs help resolve inflammation and support the remodeling processes at both tension and compression sides. This is achieved by the inhibition of osteoclastogenesis and the proinflammatory immune response mediated by effector T-helper cells, especially Th17 cells. Therefore, the balanced Th17–Treg axis is crucial for controlling bone resorption and preserving tissue integrity. Its imbalance contributes to the pathogenesis of gingivitis and periodontitis [225]. There is mutual cross-talk between Th17 cells and Tregs. Based on the local microenvironment, one subgroup may transform into another (Figure 3). Tregs often limit osteoclastogenesis by downregulating RANKL and M-CSF expression, secreting IL-10 and TGF-β, and directly interacting with osteoclast precursors. This mechanism facilitates the preservation of bone homeostasis and inhibits excessive bone resorption [226].

Orthodontic force increased Treg numbers in experimental animals. In a rat model of OTM with concurrent periodontitis, the proportion of peripheral and gingival Foxp3^+^ Tregs peaked early (approximately day 3), declined by day 7, and subsequently increased again during tooth movement. These findings indicate that the Treg response to mechanical loading in inflammatory conditions varies over time [227]. The orthodontic force applied together with corticotomy significantly increased Foxp3 expression and numbers of Tregs inside periodontal tissues both in vivo and in vivo. These processes correlated with accelerated tooth movement and decreased inflammatory bone damage [228].

Indirect evidence from models of asthma [229], RA [230], and stroke [231] indicates that PBM promotes Treg development. PDLSCs can induce Tregs, similar to other MSCs, possibly by converting pro-inflammatory Th17 cells. The shift likely appears in later OTM stages and contributes to resolution, in step with the transition from active resorption to consolidation [232,233]. In this context, PBM, via Treg-mediated immunoregulation, may enhance remodeling during OTM.

A summary of the involvement of adaptive immunity in OTM and its modulation by PBM is presented in Figure 4.

What remains unclear is whether clinical PBM protocols in OTM directly influence Th1/Th17 responses early on and Treg responses later, as well as how these effects depend on the site (compression vs. tension) and the timing of sampling. Since the available evidence on PBM is limited, heterogeneous, and dominantly indirect, the conclusions are primarily hypothesis-generating.

We assume that in the early catabolic phase, delivering higher cumulative PBM energy over broad areas may reduce the necessity for Th1 and Th17 priming. If the goal is to accelerate the process, a lower fluence with localized application may be more effective. During the transition and consolidation phases, PBM may indirectly facilitate a shift from Th17 to Tregs and help stabilize the remodeling process. In this context, site-specific application on the tension side or in anchorage regions could be worthwhile. Given that protocols are not standardized, it is essential to document and report timing, site, wavelength, irradiance, fluence, and session frequency alongside predefined biochemical and immunologic readouts. However, further prospective controlled trials are necessary to validate these assumptions.

A summary of the involvement of adaptive immune mechanisms in OTM and their modulation by PBM is presented in Table 6.

## 9. Conclusions

PBM has been used in orthodontics for years, primarily to accelerate OTM. However, the benefits have been modest, with the most consistent clinical advantage being pain reduction. Much of the observed variability likely results from non-standardized protocols for both mechanical loading and light delivery. This review systematizes current evidence of how mechanical and photonic cues are transduced into cellular messengers and integrated within shared signaling pathways. Although direct data on PBM’s impact on OTM remodeling are limited and much of the insight derives from non-skeletal models, the existing evidence is still sufficient to establish a foundational framework for integration We explore integration at multiple levels: the bidirectional crosstalk among stromal cells in the periodontal ligament and alveolar bone; the remodeling of the extracellular matrix and blood vessels; the roles of autophagy and apoptosis; and the contributions of both innate and adaptive immunity. Although interactions between force and light can be seen as either synergistic or additive, they are not consistently evident and may even diverge. For instance, mechanical force tends to promote inflammation, while PBM often alleviates these processes. Therefore, to make PBM genuinely useful in practice, more robust and extensive basic research that can be translated into well-designed clinical protocols is needed.

## 10. Future Perspectives

To effectively integrate PBM into routine orthodontic care, future studies should aim to clarify the underlying mechanisms and document patient-relevant effects. This may involve combining well-designed in vitro experiments with clear translational outcomes. Recent advancements in research models and analytical techniques support this approach. For example, single-cell RNA sequencing (RNA-seq) can identify cell-specific responses to both force and light.

Mechanically active in vitro platforms, such as two-dimensional stretching, three-dimensional compression, or force-controlled bioreactors, provide standardized loading conditions where specific wavelengths and doses can be tested for their effects on mechanotransmissive signaling. Co-culture systems that integrate PDL cells with osteoclast and osteoblast precursors, as well as immune partners like macrophages, DCs, and T cells, are also valuable. These systems may create an environment to observe cytokine dynamics, osteoimmunologic interactions, and cell-fate programs, including apoptosis and autophagy. Additionally, organoids, spheroids, and organ-on-a-chip formats may offer a more physiologically relevant context for investigating angiogenesis, ECM turnover, and the regulation of immune responses over time.

Lastly, an integrative framework that combines multi-omics, live-cell imaging, and quantitative biomarkers could assist in modeling tissue adaptation related to PBM over time. Establishing validated in vitro systems using primary human cells, controlled loading, and standardized illumination would support a more confident transition to clinical evaluation. The ultimate goal is to establish precise, practicable PBM parameters for routine orthodontic practice.

## Figures and Tables

**Figure 1 biomedicines-13-02495-f001:**
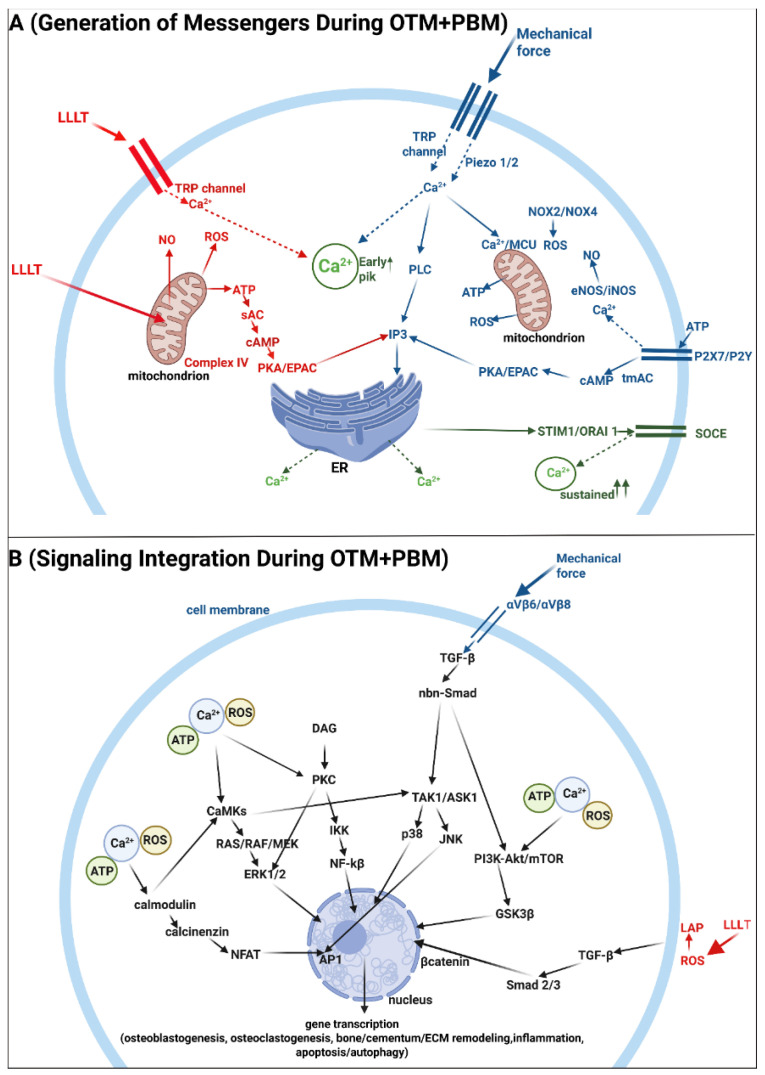
Integration of Mechanical and Photon-Induced Signals during Orthodontic Tooth Movement (OTM). (**A**) Mechanical force (MF) and photobiomodulation (PBM) generate the same core signaling intermediates (Ca^2+^, ATP, low-level ROS, NO, and cAMP) during OTM. They act through shared sensors (e.g., TRP channels) and distinct primary targets (Complex IV for PBM; Piezo1/2 for MF). PBM directly boosts mitochondrial production of ATP, low-level ROS, and NO. MF raises these signals indirectly via Ca^2+^ uptake through the mitochondrial calcium uniporter (MCU) and, predominantly, from extramitochondrial sources. The initial Ca^2+^ rise occurs through plasma-membrane Ca^2+^ channels; a sustained second wave comes from ER stores via IP_3_ generated by PLC and cAMP signaling. Note the difference in cAMP origin: MF increases cAMP downstream of extracellular ATP release, whereas PBM draws on intracellular ATP. The major subsequent Ca^2+^ influx is store-operated Ca^2+^ entry (SOCE) via STIM1/ORAI1. (**B**) MF and light converge on common signaling networks. PBM overlays mechanotransduction by supplying the same messengers, which funnel into shared hubs: calcineurin–NFAT, MAPKs (ERK, p38, JNK), PI3K–Akt–mTOR, NF-κB, TGF-β/Smad, and Wnt/β-catenin. Note the differences between TGF-β activation under MF and PBM. These interconnected pathways engage transcription factors that drive gene programs. Depending on cell target, OTM phase, and site, outputs include osteoblastogenesis, osteoclastogenesis, cell proliferation, apoptosis, autophagy, matrix remodeling, inflammation, or immune responses. Mechanical force signaling (blue); Photon light (red).

**Figure 2 biomedicines-13-02495-f002:**
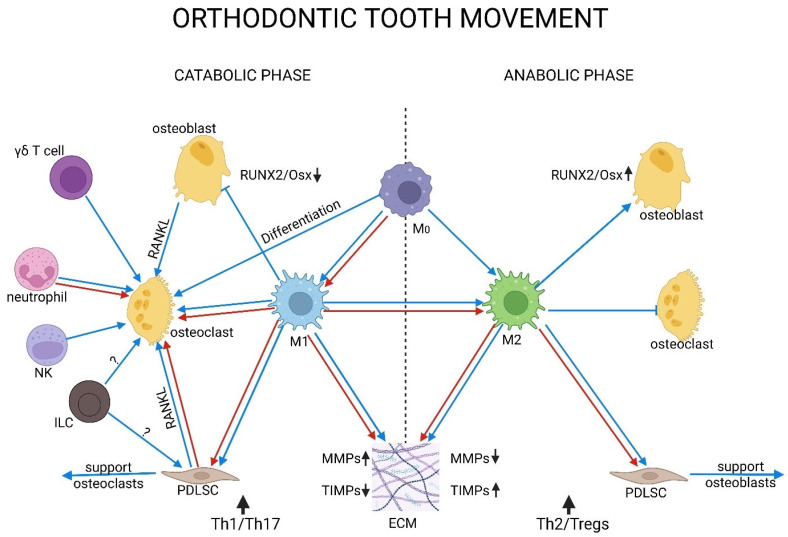
Interactions between innate immune cells and stromal components during orthodontic tooth movement (OTM) and photobiomodulation (PBM). Early (catabolic) and late (anabolic) phases of OTM often exhibit opposing M1/M2-stromal interaction patterns, both in the type and intensity of signaling. Blue lines represent mechanical force effects, while red lines indicate PBM effects. The dotted line separates OTM phases. The early inflammatory phase is marked by a rapid influx of neutrophils and monocytes into the periodontal ligament (PDL), followed by the differentiation of M1 macrophages. Cells of innate immunity generally stimulate osteoclastogenesis and osteoclast functions, either directly or through RANKL-producing cells such as osteoblasts and periodontal ligament stem cells (PDLSCs). M1 macrophages contribute to the breakdown of the extracellular matrix (ECM) while simultaneously inhibiting osteoblast activity. Photobiomodulation (PBM) may act in tandem with mechanical forces in various directions, although some inflammatory effects may vary depending on the context. During the anabolic phase, anti-inflammatory M2 macrophages become predominant. These cells, which can differentiate from M1 macrophages, promote osteoblastogenesis either directly or via PDLSCs and enhance ECM production while inhibiting osteoclasts. PBM promotes M2 differentiation and enhances the influence of M2 on PDLSCs and ECM. Th1/Th17 responses strengthen M1-dependent processes, while Th2 cells and regulatory T cells (Tregs) bolster innate immunity during the anabolic phase.

**Figure 3 biomedicines-13-02495-f003:**
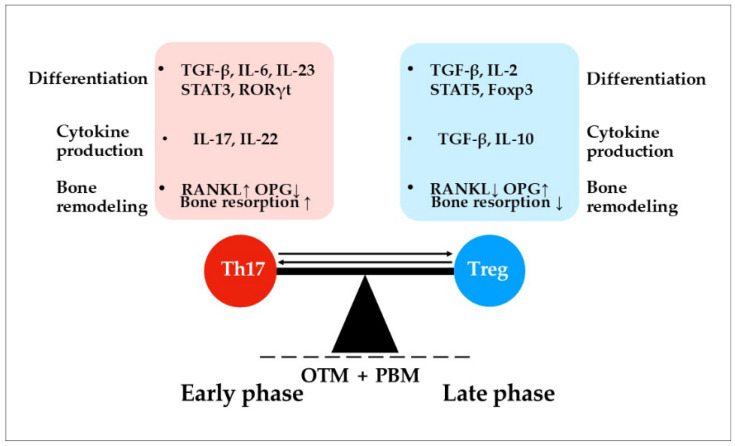
Reciprocal crosstalk between Th17 cells and regulatory T cells (Tregs) during orthodontic tooth movement (OTM) combined with photobiomodulation (PBM). The scheme shows reciprocal control and plasticity between T helper 17 (Th17) cells and regulatory T cells (Tregs): each subset inhibits the other, and, depending on the cytokine milieu and the remodeling stage, one can convert into the other. In the early OTM phase, Th17 activity dominates and promotes bone resorption by upregulating RANKL and downregulating OPG. In the later phase, Tregs prevail, limit osteoclastogenesis, and support osteoblast-driven repair, in part via reduced RANKL and increased OPG. The induction cues and secreted cytokines by these Th subsets differ. It is plausible that PBM and mechanical loading act in parallel, by enhancing Th17 and attenuating Tregs responses in the early phase, with the opposite shift during the late phase.

**Figure 4 biomedicines-13-02495-f004:**
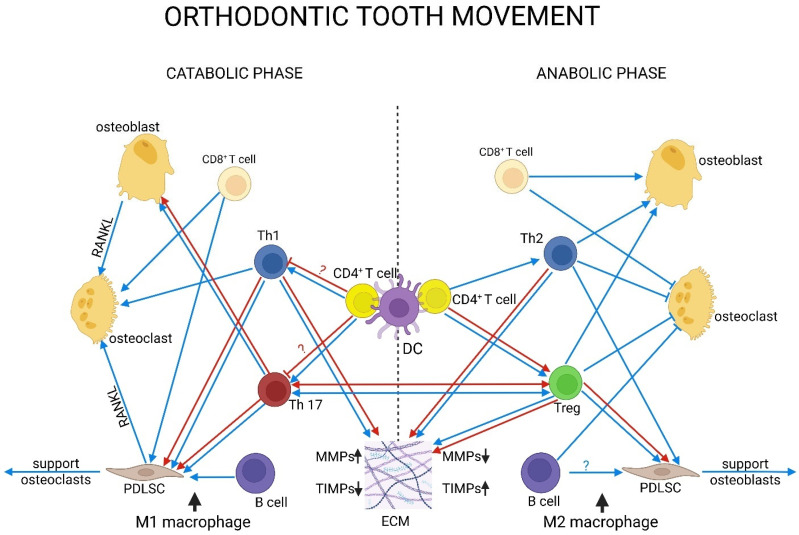
Interactions between the cells of adaptive immunity and stromal components during orthodontic tooth movement (OTM) and photobiomodulation (PBM). Early (catabolic) and late (anabolic) phases of OTM often exhibit opposing immune-stromal interaction patterns, both in the type and intensity of signaling. Blue lines represent mechanical force effects, while red lines indicate PBM effects. The T-cell response begins with the contact between damage-associated molecular pattern (DAMP)-stimulated dendritic cells (DCs) and CD4+ T cells. In the early phase of OTM, pro-inflammatory T helper 1 (Th1) and Th17 cells stimulate osteoclasts, primarily through RANKL, which is produced by periodontal ligament stem cells (PDLSCs) and osteoblasts. A similar mechanism is suggested for PBM. Mechanical force also activates B cells and CD8+ cytotoxic T cells, enhancing osteoclastogenesis. Both mechanical force and PBM promote MMP-mediated ECM degradation via Th17 and Th1 signaling. The adaptive immune response is facilitated by M1 macrophages. The late phase is marked by an increase in Th2 cells and regulatory T cells (Tregs), along with inhibitory CD8+ T cells and B cells. These cells inhibit osteoclast activity while promoting osteoblastogenesis, either directly or indirectly through PDLSCs, which develop tolerogenic properties. Tregs (differentiated from Th17 cells) and Th2 cells also contribute to ECM synthesis and neovascularization by decreasing the MMP/TIMP ratio. PBM primarily encourages Treg differentiation and enhances the effector functions of these cells, including the suppression of Th17 cells. This phase of adaptive immunity is supported by M2 macrophages. Legend: The dotted line represents the separation of two OTM phases.

**Table 2 biomedicines-13-02495-t002:** Clinical use of PBM in orthodontics.

References	Laser Types	Wavelengths (nm)	Main Indications Studied	Energy Density/Dosage (J/cm^2^)
Bakdach and Hadad (2020) [44]	Diode lasers (GaAlAs, InGaAsP)	630–940	Accelerating OTM (esp. canine retraction)	2.5–8
Inchingolo et al. (2023) [46]	Diode lasers, LED devices	630–980	Acceleration of OTM, pain reduction, miniscrew stability; root resorption	1–25
Yavagal et al. (2021) [47]	Diode lasers	670–940	Acceleration of tooth movement in children	5–25
El-Angbawi et al. (2023) [48]	Diode lasers, LED	630–940	Adjuncts for OTM, alignment, pain, treatment duration	2–20
Elgadi et al. (2023) [49]	Diode lasers	670–980	Acceleration of OTM, treatment duration reduction	0.05–108
Zheng et al. (2023) [50]	Diode lasers, intraoral/extraoral LED	635–980	Dental alignment	6–150 (cumulative dose)
Li et al. (2015) [51]	Diode lasers	630–980	Pain reduction in orthodontics	2–25 (most studies <10 J/cm^2^)
Ahmad et al. (2021) [36]	Diode, HeNe, Nd:YAG	632.8–1064	Temporomandibular joint disorders (pain, function)	1.5–112.5
Michelogiannakis et al. (2019) [52]	Diode lasers (GaAlAs)	635–830	OIIRR (root resorption)	0.01–54
Farid et al. (2019) [42]	Diode laser (InGaAs)	940	Corticotomy	5

Abbreviations: PBM—photobiomodulation; OTM—orthodontic tooth movement; LED—light-emitting diode; GaAlAs—gallium-aluminum-arsenide; InGaAsP—indium-gallium-arsenide-phosphide; InGaAs—indium-gallium-arsenide; HeNe—helium-neon; Nd:YAG—neodymium-doped yttrium aluminum garnet; OIIRR—orthodontically induced inflammatory root resorption; J/cm^2^—joules per square centimeter; nm—nanometer(s).

**Table 4 biomedicines-13-02495-t004:** Photobiomodulation and Orthodontic Tooth Movement: Integrated Cell–Cell Crosstalk, ECM Remodeling, and Vascular Adaptation.

Direction/Module	Key Mediators/Receptors	Effect in OTM (Side)	PBM Overlap/Modulation	References
Osteoblast → Osteoclast	RANKL, M-CSF, WNT5A	↑ Osteoclastogenesis (compression)	↑ RANKL, ↑ M-CSF; accelerated OTM	[16,74,103,108,109,110,111,112]
Osteoblast → Osteoclast (inhibitory)	SEMA3A, on osteoblasts → NRP1, on osteoclasts)	↓ RANKL-mediated osteoclastogenesis	PBM effect unclear	[105]
Osteoclast → Osteoblast	EFNB2 ↔ EPHB4; S1P, CTHRC1, C3; matrix-released TGF-β, IGF-1	↑ Osteoblast differentiation/activation (coupling; tension/reconstruction)	Indirectly supported via PBM effects on repair	[16,103,122]
PDLSC → Osteoclast	RANKL, PGE2 via EP4, IL-6	↑ Osteoclastogenesis (compression)	↑ RANKL; PGE2 mixed (↑ in one RCT; ↓ in another)	[16,74,114,115]
PDLSC → Osteoblast/angiogenesis	BMP/Smad, SDF-1/CXCR4, VEGF	↑ Osteogenic markers; ↑ vascular support (tension)	PBM ↑ BMP, ↑ SDF-1/CXCR4, ↑ VEGF; proliferation sometimes unchanged	[123,124,125,126,127]
ROsteocyte → Osteoclast/Osteoblast	RANKL, SOST, PGE2	SOST: ↑ RANKL (pro-OC) + ↓ Wnt (anti-OB); PGE2: pro-OC (compression)	PBM ↓ SOST (pro-OB on tension); PGE2 heterogeneous	[74,113,114,115]
ECM degradation	MMP-1/8 (fibrillar collagens), MMP-2/9/13, cathepsin K; control by TIMP-1/2	Compression: ↑ MMPs, transient ↓ TIMPs → ↑ MMP/TIMP → ↑ migration of osteoclasts/immune/endothelial cells	PBM often ↑ MMPs early; under tension, PBM may ↑ TIMPs (hypothesis: net collagen gain)	[111,116,117,118,119,128]
Adhesion/leukocyte entry	ICAM-1, VCAM-1	↑ Adhesion and transmigration of leukocytes into PDL (early, compression)	PBM may modulate via NO/VEGF; direct OTM- data limited	[120,121,122]
NO signaling	eNOS → NO	Vasodilation; modulation of osteoclasts and immune functions	PBM can ↑ eNOS/NO (mainly non-OTM models)	[120,121,122,131,132]
Angiogenesis	VEGF, HIF-1α, ERK, PI3K/Akt/mTOR	Neovascularization (tension); VEGF also ↑ MMPs (links ECM and vasculature)	PBM ↑ VEGF, ↑ HIF-1α/ERK; ↑ HUVEC proliferation/migration; ↑ vascularization in OTM model	[129,130,131,132,133,134,135]

Abbreviations: OTM—orthodontic tooth movement; PBM—photobiomodulation; PDL—periodontal ligament; PDLSC—periodontal ligament stem cell(s); ECM—extracellular matrix; OB—osteoblast; OC—osteoclast; RANKL—receptor activator of nuclear factor κB ligand; M-CSF—macrophage colony-stimulating factor; WNT5A—Wnt family member 5A; SEMA3A—semaphorin 3A; NRP1—neuropilin-1; EFNB2—ephrin-B2; EPHB4—Eph receptor B4; S1P—sphingosine-1-phosphate; CTHRC1—collagen triple helix repeat-containing 1; C3—complement component 3; TGF-β—transforming growth factor-beta; IGF-1—insulin-like growth factor-1; PGE2—prostaglandin E2; EP4—prostaglandin E receptor 4; IL-6—interleukin-6; BMP—bone morphogenetic protein; SMAD—mothers against decapentaplegic homologs (SMAD proteins); SDF-1—stromal cell-derived factor-1 (CXCL12); CXCR4—C-X-C chemokine receptor type 4; VEGF—vascular endothelial growth factor; SOST—sclerostin; MMP—matrix metalloproteinase; CTSK—cathepsin K; TIMP—tissue inhibitor of metalloproteinases; ICAM-1—intercellular adhesion molecule-1; VCAM-1—vascular cell adhesion molecule-1; eNOS—endothelial nitric oxide synthase; NO—nitric oxide; HIF-1α—hypoxia-inducible factor-1 alpha; ERK—extracellular signal-regulated kinase; PI3K/Akt/mTOR—phosphoinositide 3-kinase/protein kinase B/mechanistic target of rapamycin pathway; HUVEC(s)—human umbilical vein endothelial cell(s); Legend: ↑ increase; ↓ decrease; → mark dirrection; tension = tension side; compression = compression side.

**Table 5 biomedicines-13-02495-t005:** The Effect of M1 and M2 Macrophages on Osteoblasts, Osteoclasts, PDLSCs, and ECM During Combined Orthodontic Tooth Movement and Photobiomodulation.

M1/M2	Target	Key Mediators	Dominant Paths in Target	Functional Effects	OTM Net Effect	References
M1	Osteoblasts	TNF-α, IL-1β, IL-6, IFN-γ; **NO/ROS**; EV (miR-155)	↑**NF-κB**, **p38/JNK**; ↓**BMP/Smad**; ↓**Wnt/β-catn** (↑DKK1/SOST)	↓RUNX2/OSX, ↓ALP, ↓COL1A1/OCN, ↓mineralization; ↑RANKL, ↓OPG. PBM: early transient M1 priming; later shift to M2.	Inhibits bone formation; shifts toward resorption	[89,171,172]; PBM: [185,186,187,188]
M1	Osteoclasts (via stroma)	TNF-α, IL-1β, **PGE_2_**; induces **RANKL** in osteoblasts/PDLSCs	↑RANK–**RANKL**–**NF-κB** in precursors	↑Differentiation and resorptive activity. PBM: may add to early RANKL-driven OC-genesis.	Boosts resorption (early compression side)	[172,177,191,192,193,194]; PBM: [186,187,188]
M1	PDLSCs	TNF-α, IL-1β, IL-6; **NO/ROS**; EV (miR-155)	↑**NF-κB**/p38; ↓**BMP/Smad**; ↓**Wnt/β-catn**	↓Proliferation/migration; ↓osteogenesis; ↑RANKL/↓OPG. PBM: can down-tune sustained M1 signaling over time.	Pro-resorptive, anti-repair microenvironment	[180,182,194]; PBM: [186,187,188]
M1	ECM/vasculature	TNF-α, IL-1β; ↑MMP-1/3/9; ↓TIMPs; early ↓**VEGF**	—	Matrix degradation > deposition; delayed angiogenesis. PBM: context-dependent MMP/TIMP–VEGF effects.	Fast breakdown; slower consolidation	[116,117,118,119,129,130,131,132,133,134,135,194,195]
M2	Osteoblasts	IL-10, TGF-β, IGF-1, PDGF; pro-osteogenic EV/miRNA	↓**NF-κB**; ↓**MAPK**; ↑**BMP/Smad**; ↑**PI3K–Akt/ERK**; ↑Wnt/β-catenin; ↑**NFATc**	↑RUNX2/OSX, ↑ALP, ↑COL1A1/OCN, ↑mineralization; ↓RANKL, ↑OPG. PBM: supports M2 polarization.	Promotes bone formation (repair phase)	[89,171,182,183]; PBM: [186,187,188,189,190]
M2	Osteoclasts (via stroma)	IL-10, TGF-β; ↑**OPG**/↓**RANKL** in stroma	↓**NF-κB**; ↓**MAPK**; ↑**NFATc**; weakened RANK–RANKL signaling	↓Osteoclastogenesis and activity. PBM: M2 shift restrains resorption.	Restrains resorption; favors balance	[172,182,183]; PBM: [186,187,188]
M2	PDLSCs	IL-10, TGF-β, **VEGF**, PDGF; reparative EV/exosomes	↓**NF-κB**; ↑**BMP/Smad**; ↑**PI3K–Akt/ERK**; ↑**Wnt/β-catn**.	↑Proliferation/migration; ↑osteogenesis & mineralization; ↑OPG/↓RANKL. PBM: pro-repair setting (tension).	Pro-repair, pro-osteogenic setting (tension side)	[182]; PBM/angiogenesis: [123,124,125,126,127,129,130,131,132,133,134,135]
M2	ECM/vasculature	TGF-β; early ↑MMP-2/9 then ↑TIMPs; ↑**VEGF**	—	Controlled early remodeling; later ↑collagen I; enhanced angiogenesis. PBM: ↑VEGF; rebalances MMP/TIMP.	Organized remodeling and stable consolidation	[129,130,131,132,133,134,135,194]; PBM angiogenesis: [129,130,131,132,133,134,135,188,189,190]

Mediators/pathways shared by OTM and PBM are highlighted in bold. Abbreviations: M1—classically activated (pro-inflammatory) macrophage; M2—alternatively activated (pro-resolving) macrophage; PBM—photobiomodulation; OTM—orthodontic tooth movement; OB—osteoblast; OC—osteoclast; PDLSC(s)—periodontal ligament stem cell(s); ECM—extracellular matrix; TNF-α—tumor necrosis factor-alpha; IL-1β—interleukin-1 beta; IL-6—interleukin-6; IFN-γ—interferon-gamma; NO—nitric oxide; ROS—reactive oxygen species; EV—extracellular vesicle(s); miR-155—microRNA-155; NF-κB—nuclear factor kappa-B; p38—p38 mitogen-activated protein kinase; JNK—c-Jun N-terminal kinase; MAPK—mitogen-activated protein kinase; BMP—bone morphogenetic protein; Smad—Smad signal transducers; Wnt/β-catenin—Wnt/β-catenin pathway; DKK1—Dickkopf-related protein 1; SOST—sclerostin; RUNX2—Runt-related transcription factor 2; OSX—Osterix (SP7); ALP—alkaline phosphatase; COL1A1—collagen type I alpha 1 chain; OCN—osteocalcin (BGLAP); RANK—receptor activator of NF-κB; RANKL—receptor activator of NF-κB ligand; OPG—osteoprotegerin; PGE_2_—prostaglandin E_2_; TIMP(s)—tissue inhibitor(s) of metalloproteinases; MMP-1/3/9—matrix metalloproteinases-1/3/9; VEGF—vascular endothelial growth factor; IGF-1—insulin-like growth factor-1; PDGF—platelet-derived growth factor; PI3K—phosphoinositide 3-kinase; Akt—protein kinase B; ERK—extracellular signal-regulated kinase; NFATc—nuclear factor of activated T-cells (cytoplasmic component); OC-genesis—osteoclastogenesis. ↑ increase; ↓ decrease.

**Table 6 biomedicines-13-02495-t006:** Adaptive Immunity in Orthodontic Tooth Movement (OTM) with Photobiomodulation (PBM): Shared Signals, Targets, and Combined Functional Effects.

Cell Type	Site	Shared Signals (PBM+OTM)	Main Targets	Functional Effects (Combined)	Parameters	References
Th1	Compress.	**IFN-γ**, TNF; stromal **RANKL**/**OPG**; **NF-κB**	Osteoclast precursors; PDL stroma	↑RANKL/↓OPG → ↑osteoclasts; ↑ECM degradation (MMP-9↑/TIMP↓). PBM may act synergistically early via inflammatory cues; context-dependent bone effects via IFN-γ.	**RANKL**/**OPG,**TRAF6–**NF-κB, MMP-9**, TRAP; PBM: JAK2/STAT4/STAT5	[2,184,185,186,187,188,189,200,201,202]
Th1	Tension	**IL-10** (resolution), pro-repair angiogenic (**VEGF**)	Osteoblast lineage; microvasculature	↑Osteogenesis (↑RUNX2/ALP); ↑angiogenesis (VEGF↑); matrix deposition.	VEGF, RUNX2, ALP, OCN; PBM: JAK2/STAT4/STAT5	[201,202]
Th17	Compress.	IL-17A → **RANKL**/**OPG**; **NF-κB**	Osteoclast precursors; stroma	OTM: ↑RANKL/↓OPG → ↑osteoclasts; MMP-9↑/TIMP↓. PBM (general) may later temper excessive inflammation.	**RANKL**/**OPG, NF-κB**, **MMP-9**, TRAP	[203,204,205,206,207]
Th17	Tension	Reduced Th17 tone; pro-repair **VEGF**	Osteoblasts; endothelium	↑Osteogenesis (↑RUNX2/ALP); ↑angiogenesis (VEGF↑).	**VEGF**, **RUNX2**, **ALP**	[203,205]
Th2	Compress.	**IL-10**; lowered **NF-κB**; tempered **RANKL**/**OPG** drive	Stroma; osteoclast lineage	↓RANKL signaling; restrained osteoclasts; MMP-9↓/TIMP↑.	**IL-10**, **NF-κB**, **MMP-9**, TRAP	[208,209]
Th2	Tension	IL-10, **VEGF**	Osteoblasts; vessels	↑Osteogenesis (↑RUNX2/ALP); ↑angiogenesis (VEGF↑); organized ECM.	**VEGF**, RUNX2, ALP	[208,209]
Tregs	Compress.	**IL-10**, **TGF-β**; **NF-κB** suppression	Osteoclast precursors; stroma; macrophages	↓RANKL/↑OPG; restrain osteoclasts; limit MMPs; facilitate resolution.	**IL-10**, **RANKL**/**OPG, NF-κB**, TRAP↓	[222,223,224,225,226]
Tregs	Tension	**IL-10**; repair cues incl. **VEGF**	Osteoblasts; endothelium	↑Osteogenesis (↑RUNX2/ALP); ↑angiogenesis (VEGF↑). PBM likely promotes Treg bias in later OTM stages.	**VEGF**, **RUNX2**, **ALP**	[222,223,224,225,226,230,231]
CD8^+^	Compress.	IFN-γ, RANKL/OPG, NF-κB	Osteoclast precursors; stroma	↑RANKL/↓OPG → ↑osteoclasts; MMP-9↑/TIMP↓; OIIRR risk. FoxP3^+^ CD8^+^ can counter via IFN-γ/CTLA-4.	IFN-γ, RANKL/OPG, NF-κB, MMP-9	[210,211]
CD8^+^	Tension	Regulatory skew **(IL-10),** pro-repair VEGF	Osteoblasts; vessels	↑Osteogenesis (↑RUNX2/ALP); ↑angiogenesis (VEGF↑).	VEGF, RUNX2, ALP	[210,211]
B cells	Compress.	Activated B → **RANKL**; Bregs → **OPG**, **IL-10**; **NF-κB** balance	Osteoclast precursors; stroma	Context-dependent: ↑RANKL (activation) vs. ↑OPG/IL-10 (restraint); MMP-9 tempered with Breg bias.	**RANKL**/**OPG, IL-10**, **MMP-9**, TRAP	[212,213,214,215]
B cells	Tension	**IL-10**/**OPG** support; **VEGF**	Osteoblasts; endothelium	Matrix deposition; ↑angiogenesis (VEGF↑); consolidation.	**OPG**, **VEGF**, **RUNX2**, **ALP**	[214,215]

Signals/parameters shared by OTM and PBM are highlighted in bold. Abbreviations: OTM—orthodontic tooth movement; PBM—photobiomodulation; PDL—periodontal ligament; PDLSC—periodontal ligament stem cell; ECM—extracellular matrix; Th—T helper cell; Tregs—regulatory T cells; CD8^+^—cytotoxic T lymphocyte; Bregs—regulatory B cells; IFN-γ—interferon-gamma; TNF– tumor necrosis factor; IL—interleukin; TGF-β—transforming growth factor-beta; NF-κB—nuclear factor kappa-B; TRAF6—TNF receptor-associated factor 6; RANK—receptor activator of NF-κB; RANKL—receptor activator of NF-κB ligand; OPG—osteoprotegerin; MMP—matrix metalloproteinase; TIMP—tissue inhibitor of metalloproteinases; TRAP—tartrate-resistant acid phosphatase; RUNX2—runt-related transcription factor 2; ALP—alkaline phosphatase; OCN—osteocalcin; VEGF—vascular endothelial growth factor; CTLA-4—cytotoxic T-lymphocyte–associated protein 4; FoxP3—forkhead box P3; OIIRR—orthodontically induced inflammatory root resorption; PGE_2_—prostaglandin E_2_. Legend: ↑ increase; ↓ decrease; → mark dirrection.

## Data Availability

Not applicable.

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
