# Peer review of "Photobiomodulation Meets Mechanotransduction: Immune-Stromal Crosstalk in Orthodontic Remodeling"

_biomedicines, 2025, doi:10.3390/biomedicines13102495_

Round 1
Reviewer 1 Report
Comments and Suggestions for Authors
1 . Abstract
- The abstract is somewhat lengthy and descriptive. It would benefit from a more concise formulation of the main message. In particular, the concluding sentences should more clearly emphasize that LLLT functions not only as an adjunctive tool but also as a “mechanical–photonic integrator.”
- Figures, Tables
- The manuscript is primarily text-based and lacks sufficient visual summaries.
→ Suggested additions: a schematic illustration of key signaling pathways (MAPK, PI3K/Akt, Wnt/β-catenin cross-talk), and a summary table outlining the effects of LLLT at different stages of orthodontic tooth movement (OTM). - Figure legends should be concise and self-explanatory so that figures can be understood independently.
- Terminology Consistency
- The terms “PBM,” “LLLT,” and “photobiomodulation” are used interchangeably. It would be clearer to define the terminology at the beginning and then use one term consistently throughout the manuscript.
- Likewise, “mechanotransduction” and “mechanosensing” appear to be used without distinction in some places. More precise and consistent usage of these terms is recommended.
- Immune Section
- The section on immune cells is comprehensive but tends to combine orthodontic-specific findings with general immunological descriptions. Distinguishing between direct evidence (from orthodontic models) and indirect evidence (from broader immunology) would improve clarity.
- The subsection on adaptive immunity, particularly the Th17/Treg balance, is highly relevant and could be further strengthened by the inclusion of a figure or summary table to improve readability and impact.
Author Response
Response to Reviewer 1
- Abstract
Comment: The abstract is somewhat lengthy and descriptive. It would benefit from a more concise formulation of the main message. In particular, the concluding sentences should more clearly emphasize that LLLT functions not only as an adjunctive tool but also as a “mechanical–photonic integrator.”
Response: Thank you for this suggestion. The abstract has been shortened and reformulated to foreground the central message. The concluding sentences now explicitly state that photobiomodulation (PBM/LLLT) functions not only as an adjunct but also as an integrator of mechanical and photonic signals.
- Figures and Tables
Comment: The manuscript is primarily text-based and lacks sufficient visual summaries. Suggested additions: a schematic illustration of key signaling pathways (MAPK, PI3K/Akt, Wnt/β-catenin cross-talk) and a summary table outlining the effects of LLLT at different stages of orthodontic tooth movement (OTM).
Response: We appreciate this recommendation. We have added a comprehensive schematic (Figure 1) that depicts how mechanical force and PBM generate shared messengers and converge on common signaling hubs (e.g., Ca²⁺–calcineurin–NFAT; MAPK; PI3K–Akt–mTOR; NF-κB; TGF-β/Smad; Wnt/β-catenin). A concise, self-contained legend accompanies the figure to ensure stand-alone interpretability.
Regarding the proposed summary table: at the request of the second reviewer, we have already added two new tables regarding clinical application and summarizing PBM-related outcomes and parameters. To avoid redundancy and keep the manuscript streamlined, we did not include an additional table here. Instead, each relevant section now ends with brief, focused take-home points that specify PBM’s effects at the corresponding OTM stage.
Comment: Figure legends should be concise and self-explanatory so that figures can be understood independently.
Response: Implemented. The legends for all three figures are written to be concise and self-explanatory, allowing each figure to be understood without referring back to the main text.
- Terminology Consistency
Comment: The terms “PBM,” “LLLT,” and “photobiomodulation” are used interchangeably. It would be clearer to define the terminology at the beginning and then use one term consistently throughout the manuscript.
Response: Thank you for this important point. In the Introduction, we now define “photobiomodulation (PBM)” as the umbrella term encompassing low-level laser therapy (LLLT) and light-emitting diode (LED) applications. Throughout the manuscript we use PBM consistently as the general term, while LLLT is retained only when citing studies that explicitly employed lasers.
Comment: “Mechanotransduction” and “mechanosensing” appear to be used without distinction in some places. More precise and consistent usage is recommended.
Response: Corrected. We now reserve “mechanosensing” for stimulus detection at the membrane/organelle level and “mechanotransduction” for the downstream conversion into biochemical signals. All occurrences were reviewed and harmonized accordingly.
- Immune Section
Comment: The section on immune cells is comprehensive but tends to combine orthodontic-specific findings with general immunological descriptions. Distinguishing between direct evidence (from orthodontic models) and indirect evidence (from broader immunology) would improve clarity.
Response: We agree. The section has been shortened, reorganized, and clearly partitioned into (i) direct OTM evidence and (ii) contextual immunology that could explain plausible PBM effects. Where appropriate (at the end of subsections; marked in red), we indicate whether PBM effects are demonstrated, supported by related models, or hypothesized for OTM.
Comment: The subsection on adaptive immunity, particularly the Th17/Treg balance, is highly relevant and could be strengthened by a figure or summary table.
Response: Implemented. We have added Figure 2, which illustrates the Th17/Treg balance and key cytokine nodes relevant to compression vs. tension contexts. The legend explains the crosstalk and expected PBM-modulated checkpoints to aid readability and impact.
Reviewer 2 Report
Comments and Suggestions for Authors
I would like to thank the Authors for their effort in preparing this comprehensive and scientifically relevant manuscript. The topic is timely and of interest to both clinicians and researchers, as it addresses the interplay between mechanotransduction, immune dynamics, and photobiomodulation in orthodontic tooth movement.
However, in its present form the manuscript is excessively long, sometimes redundant, and would benefit from substantial condensation and clearer synthesis of the key findings. For these reasons, I recommend major revision.
Specific comments and detailed suggestions for improvement are provided in the annotated PDF attached to this review.

The English language is generally adequate and understandable; however, several sentences are overly long and complex, which sometimes reduces clarity. A careful language editing and stylistic revision would improve readability and flow, making the manuscript more concise and easier to follow.
Author Response
Response to Reviewer 2
Comments on the Quality of English Language
Comment:
The English language is generally adequate and understandable; however, several sentences are overly long and complex, which sometimes reduces clarity. A careful language editing and stylistic revision would improve readability and flow, making the manuscript more concise and easier to follow.
Response: Thank you for this suggestion. While the original grammar was correct, we agree that some sentences were long and complex. We have performed stylistic editing to shorten sentences, improve readability, and streamline the flow. Grammar has been re-checked and remains correct.
General Assessment
Comment:
I would like to thank the Editors of Biomedicines for the opportunity to review this comprehensive manuscript entitled “Photobiomodulation Meets Mechanotransduction: Immune and Cellular Cross Talk during Orthodontic Tooth Movement”. I also congratulate the Authors for their effort in compiling a broad and detailed review that addresses both mechanobiological and immunological aspects of orthodontic tooth movement (OTM) and integrates them with photobiomodulation (PBM). The manuscript is timely, scientifically relevant, and could represent a useful reference for clinicians and researchers in orthodontics, immunology, and laser therapy. Nevertheless, the paper is currently very long and sometimes redundant, with an excess of highly technical details that may reduce readability for the general readership of the journal. Some sections would benefit from condensation, improved critical synthesis, and a more consistent balance between descriptive and interpretative content. For these reasons, I recommend major revision before the manuscript can be considered for publication.
Response: Thank you for this constructive overview. In the revised version, the manuscript has been shortened and reorganized to reduce redundancy. Sections were edited for conciseness, technical details were streamlined, and greater emphasis was placed on critical synthesis and interpretation.
Title & Abstract
Comment:
• The title is clear and reflects the content, but could be made more concise.
Response: Thank you. The title has been shortened to:
“Photobiomodulation Meets Mechanotransduction: Immune–Stromal Crosstalk in Orthodontic Remodeling.”
Comment:
• The abstract is informative but overly dense. It summarizes well the biological rationale, yet it should be more explicit in stating the key conclusions and gaps in evidence. At present, it reads almost as a mini-review rather than a focused summary. A clearer take-home message is needed.
Response: Thank you. The abstract has been rewritten and made more concise. It now states the key conclusions and highlights the main evidence gaps, incorporating your suggestions as well as those from the other reviewer.
Introduction
Comment:
• Well-written and provides an adequate overview of OTM and PBM.
• However, it is quite lengthy. The introduction could be streamlined to emphasize the novelty of this review (the immune and cellular cross-talk under PBM during OTM) rather than reiterating basic concepts of orthodontic biology extensively covered in previous literature.
Response: Thank you. The Introduction has been shortened and refocused to emphasize the novelty of the review, especially immune and cellular crosstalk under PBM during OTM. We also clarified terminology at the outset, defining photobiomodulation (PBM) as the umbrella term and retaining LLLT only where specifically used by cited authors. These adjustments incorporate your suggestions.
Sections 2–3 (Key Cellular Players; Biological and Molecular Basis of OTM)
Comment:
• These sections are thorough but somewhat excessive in detail, especially for a review. The text resembles a textbook chapter.
Response: Thank you for the constructive comments. The former Section 2 has been deleted. The former Section 3 is now Section 2, rewritten in a more concise form. We decided to keep it as a brief framework to introduce subsequent sections related to the combined PBM + OTM outcomes.
Comment:
• I suggest condensing molecular pathways (e.g., MAPK, PI3K/Akt, Wnt) and directing the reader to schematic figures or tables summarizing key interactions.
Response: The description of signaling pathways was condensed in this section, but extended in Section 5, where a summary schematic has been added.
Comment:
• Greater emphasis should be placed on how this biology directly connects to PBM, since otherwise these sections appear only tangentially linked to the central topic.
Response: This is directed to other sections, where the effect of PBM was evaluated. As mentioned above, we kept this section in a reduced form to serve as an introductory foundation for the PBM-focused analyses that follow.
Section 4 (Photobiomodulation in Dentistry)
Comment:
• This section is relevant and informative but mixes general dental applications with orthodontics.
• It could be shortened by focusing on orthodontic outcomes and briefly citing broader dental uses only as background.
• A table summarizing laser types, wavelengths, and main indications would greatly improve clarity.
Response: Thank you for the comment. This section has been shortened and now appears as Section 3. It provides a brief background on general laser use in dentistry, and we added Table 1 summarizing laser types, wavelengths, key irradiation parameters, and principal clinical outcomes (synthesized from 10 references), while keeping the focus on orthodontic relevance.
Section 5 (Systematic Reviews on LLLT in Orthodontics)
Comment:
• This is a strength of the paper, but the synthesis needs improvement.
• Many systematic reviews and meta-analyses are cited in a descriptive manner; the reader would benefit from a comparative summary highlighting consistent findings, discrepancies, methodological flaws, and gaps.
• Consider including a table summarizing the systematic reviews (author/year, number of studies, main findings, limitations).
Response: Thank you for this critique. Initially, we wanted to present summaries from large systematic studies and meta-analyses, because hundreds of individual trials were reported and extensively reviewed in other papers. According to your suggestions, this section has been condensed and reorganized to present a clearer comparative synthesis. We added a summary table (now Table 2) covering 20 systematic reviews/meta-analyses (author/year, number of studies, main findings, limitations). The text now emphasizes unified conclusions, key discrepancies, methodological gaps, and practical suggestions for future studies.
Sections 6–9 (Molecular Aspects, Cross-talk, Matrix Remodeling, Apoptosis/Autophagy)
Comment:
• These sections are very dense, with extensive mechanistic details. While informative, they risk overwhelming the reader.
• The novelty here should be to highlight how PBM modulates each pathway, not simply to restate the biology of OTM.
• The narrative would be more effective if structured around a critical question: “What is known about PBM’s influence on this pathway/cell type, what is contradictory, and what remains unclear?”
• Apoptosis/autophagy section is particularly interesting but needs clearer linkage to orthodontic clinical implications.
Response: Thank you for your comments. In the revised manuscript, each section has been rewritten, shortened, and refocused. Each section now ends with a concise synthesis addressing what is known, what is contradictory, and what remains unclear regarding PBM (marked in red at the end of each of these sections, including our hypothesises). The previous Section 6 (now Section 5) is now extended in certain aspects to make it clearer how mechanical force and photon energy generate shared messengers and converge on common signaling pathways; this is illustrated by Figure 1. In addition, the apoptosis/autophagy part now more directly links mechanisms to orthodontic clinical implications (added at the end of both subsections in red).
Section 10 (Immune Dynamics)
Comment:
• This is a highly original contribution and sets the paper apart from existing reviews.
• However, the text is excessively long and at times digresses into immunology not strictly related to orthodontics.
• I recommend condensing, providing clear figures summarizing innate and adaptive immune responses under PBM, and explicitly stating the clinical translation (e.g., “This may suggest PBM can accelerate tooth movement by enhancing early pro-inflammatory responses, but evidence is still inconclusive”).
Response: Thank you for these remarks. The section has been shortened; foundational immunology is kept minimal to support orthodontic-specific content. PBM effects are more clearly highlighted, with a note that many results originate outside the OTM context. At the end of the innate and adaptive subsections, we explicitly state the clinical translation and the need for caution in extrapolation (marked in red). We added Figure 2 (Th17/Treg balance; asked by another reviewer) and retained our original immune network schematic (now Figure 3) with a simplified legend to clarify interactions. In our opinion, no further changes are necessary.
Figures & Tables (overall)
Comment:
• Currently, the text is very heavy. More visual summaries (schematic diagrams, tables of systematic reviews, flowcharts of immune pathways) would significantly improve readability.
Response: Thank you for these suggestions. The revised manuscript now includes three figures (two are new) and two new tables, each figure accompanied by concise, self-contained legends that highlight the key points.
Conclusions
Comment:
• The conclusions are somewhat buried under the wealth of details.
• This section should be expanded to provide a clear synthesis: What is the current evidence? What are the main controversies? What should future research address?
• Without a sharper conclusion, the reader is left without a clear final message.
Response: We agree. The Conclusion has been revised to deliver a clearer synthesis of current evidence, the main controversies, and explicit priorities for future research. In addition, Future Perspectives are now more prominently highlighted.
Round 2
Reviewer 2 Report
Comments and Suggestions for Authors
I would like to thank the authors for their thorough revision of the manuscript, which is now significantly improved and more focused. I recommend minor revision before acceptance. The remaining suggestions for further improvement are detailed in the annotated PDF attached.

Author Response
Response to Reviewer (Second Revision)
First, we would like to thank you once again for your constructive suggestions. This revision process has taken a long time due to the initially different concept for writing this review. Initially, our intention was to present this narrative review in several sections, written so that each section could be read as a separate chapter. This was the reason why there were perhaps some repetitions. Once we agreed that it should now be a single, unified manuscript, we removed Section 2 as redundant and have done everything requested by both reviewers. However, it seems that some instructions were not fully clear to us, which may have led to a mismatch between our interpretation and your expectations. In addition, my personal view (M.Č.) is that every table should be accompanied by a clear textual explanation, whereas your preference is to keep the text to a minimum and leave the details in the table. These are quite different concepts. To reach a compromise, we adopted your suggestions in several segments and retained our original concept in others; these choices are explained point by point. These are, of course, the sections that contribute most to the originality of the article.
Point- by- point responses
Abstract: Much improved, but could still end with a more definitive clinical translation (e.g., “At present, PBM remains a promising but not yet standardized adjunct to orthodontics”).
Reply (Abstract): We accepted this suggestion and added a similar sentence (marked in red at the end of the abstract).
Introduction: Well balanced now, though one or two paragraphs could be cut without loss of content.
Reply (Introduction): We omitted several sentences, so the Introduction is now maximally condensed. Some sentences were rephrased for clarity.
Sections 2–3: Condense further the description of signaling pathways; consider directing readers to figures/tables instead of lengthy text.
Reply (Sections 2–3): The previous Section 2 has already been omitted. Section 3 (now Section 2) is maximally condensed, restructured, and supplemented with Table 1 showing the activation of key transcription factors during orthodontic tooth movement (OTM)—receptors, intermediary molecules, signaling pathways, and resulting outcomes.
Sections 6–9: Still overly dense. Please highlight PBM-specific findings more clearly and reduce general descriptions of molecular cascades.
Reply (Sections 6–9):
Section 5 (previous Section 6) was not significantly modified, except that some unnecessary sentences were removed (these remain struck through in the text). New sentences are marked in red. We consider this a core section of the review and the signaling content is already maximally simplified. In addition, Figure 1 is very illustrative and easy to follow, with an explanatory legend that allows reading without the main text.
Section 6 was reconstructed from the previous Sections 7 and 8. Instead of listing PBM effects after mechanical-force effects, PBM is now integrated at the end of each completed subsection. This section is considerably reduced. The part on imRs was removed. Unknowns are highlighted, gaps are identified, and we provide our view of what these findings and hypotheses mean for clinical practice. Intercellular communication and ECM/vascular interactions are presented in the newly added Table 4.
Section 7 (previous Section 9) was reconstructed: autophagy is presented first, followed by apoptosis. We propose our hypothesis on how PBM modulation could improve OTM outcomes through effects on apoptosis and autophagy. The limited amount of PBM-specific data did not justify adding another table. Since two tables and two new figures have already been added, an additional low-information table from this section would be redundant.
Section 10: Condense immunological details that are not directly connected to OTM under PBM. Provide a clearer figure summarizing innate vs adaptive immune effects.
Reply (Section 8, previous section 10): We omitted basic background details (in fact, only two to three sentences describe this background). Some parts were condensed and some clarified. Deletions remain struck through; new text is marked in red. The main change is that the previous, more complex figure was split into two (innate immunity and adaptive immunity). These two figures (Figure 2 and Figure 4) are maximally simplified and further explained in their legends. Points that are not included in figures or text are additionally illustrated in Tables 5 and 6, which show mediators and signaling pathways that are either specific to certain energies or shared between OTM and PBM (marked in bold).
Figures: Simplify schematic diagrams to enhance clarity for a broader readership.
Reply (Figures): As explained above, all four figures are now simpler, clearer, and well explained in the legends. All abbreviations are provided below the figures and below the tables.
Conclusions: Strengthen the clinical message by more clearly distinguishing well-supported evidence (pain reduction, modest acceleration) from speculative hypotheses (long-term immune effects).
Reply (Conclusions): The Conclusion is now highly condensed. It clearly states the clinical message and emphasizes the importance of basic research to support better standardization of combined PBM+OTM therapy.